# Fast, accurate antibody structure prediction from deep learning on massive set of natural antibodies

Jeffrey A. Ruffolo [1], Lee-Shin Chu [2], Sai Pooja Mahajan[2] & Jeffrey J. Gray [1,2] ✉

Antibodies have the capacity to bind a diverse set of antigens, and they have become critical therapeutics and diagnostic molecules. The binding of antibodies is facilitated by a set of six hypervariable loops that are diversified through genetic recombination and mutation. Even with recent advances, accurate structural prediction of these loops remains a challenge. Here, we present IgFold, a fast deep learning method for antibody structure prediction. IgFold consists of a pre-trained language model trained on 558 million natural antibody sequences followed by graph networks that directly predict backbone atom coordinates. IgFold predicts structures of similar or better quality than alternative methods (including AlphaFold) in significantly less time (under 25 s). Accurate structure prediction on this timescale makes possible avenues of investigation that were previously infeasible. As a demonstration of IgFold's capabilities, we predicted structures for 1.4 million paired antibody sequences, providing structural insights to 500-fold more antibodies than have experimentally determined structures.

Antibodies play a critical role in the immune response against foreign pathogens. Through genetic recombination and hyper-mutation, the adaptive immune system is capable of generating a vast number of potential antibodies. Immune repertoire sequencing provides a glimpse into an individual's antibody population[1]. Analysis of these repertoires can further our understanding of the adaptive immune response[2] and even suggest potential therapeutics[3]. However, sequence data alone provides only a partial view into the immune repertoire. The interactions that facilitate antigen binding are determined by the structure of a set of six loops that make up a complementarity determining region (CDR). Accurate modeling of these CDR loops provides insights into these binding mechanisms and promises to enable rational design of specific antibodies[4]. Five of the CDR loops tend to adopt canonical folds that can be predicted effectively by sequence similarity[5]. However, the third CDR loop of the heavy chain (CDR H3) has proven a challenge to model due to its increased diversity, both in sequence and length[6,7]. Further, the position of the H3 loop at the interface between the heavy and light chains makes its conformation dependent on the inter-chain orientation[8,9]. Given its

central role in binding, advances in prediction of H3 loop structures are critical for understanding antibody-antigen interactions and enabling rational design of antibodies.

Deep learning methods have brought about a revolution in protein structure prediction[10,11]. With the development of Alpha-Fold, accurate protein structure prediction has largely become accessible to all[12]. Beyond monomeric proteins, AlphaFold-Multimer has demonstrated an impressive ability to model protein complexes[13]. However, performance on antibody structures remains to be extensively validated. Meanwhile, antibody-specific deep learning methods such as DeepAb[14] and ABlooper[15] have significantly improved CDR loop modeling accuracy, including for the challenging CDR H3 loop[7,16]. DeepAb predicts a set of inter-residue geometric constraints that are fed to Rosetta to produce a complete $F_V$ structure[14]. ABlooper predicts CDR loop structures in an end-to-end fashion, with some post-prediction refinement required, while also providing an estimate of loop quality[15]. Another tool, NanoNet[17], has been trained specifically for prediction of single-chain antibodies (nanobodies) and provides fast predictions. While effective, certain

[1]Program in Molecular Biophysics, The Johns Hopkins University, Baltimore, MD 21218, USA. [2]Department of Chemical and Biomolecular Engineering, The Johns Hopkins University, Baltimore, MD 21218, USA. ✉e-mail: jgray@jhu.edu

design decisions limit the utility of both models. DeepAb predictions are relatively slow (10 min per sequence), cannot effectively incorporate template data, and offer little insight into expected quality. ABlooper, while faster and more informative, relies on external tools for framework modeling, cannot incorporate CDR loop templates, and does not support nanobody modeling.

Concurrent with advances in structure prediction, self-supervised learning on massive sets of unlabeled protein sequences has shown remarkable utility across protein modeling tasks[18,19]. Embeddings from transformer encoder models trained for masked language modeling have been used for variant prediction[20], evolutionary analysis[21,22], and as features for protein structure prediction[23,24]. Auto-regressive transformer models have been used to generate functional proteins entirely from sequence learning[25]. The wealth of immune repertoire data provided by sequencing experiments has enabled development of antibody-specific language models. Models trained for masked language modeling have been shown to learn meaningful representations of immune repertoire sequences[22,26,27], and even repurposed to humanize antibodies[28]. Generative models trained on sequence infilling have been shown to generate high-quality antibody libraries[29,30].

In this work, we present IgFold: a fast, accurate model for end-to-end prediction of antibody structures from sequence. IgFold leverages embeddings from AntiBERTy[22], a language model pre-trained on 558 million natural antibody sequences, to directly predict the atomic coordinates that define the antibody structure. Predictions from IgFold match the accuracy of the recent AlphaFold models[10,13] while being much faster (under 25 s). IgFold also provides flexibility beyond the capabilities of alternative antibody-specific models, including robust incorporation of template structures and support for nanobody modeling.

## Results

### End-to-end prediction of antibody structure

Our method for antibody structure prediction, IgFold, utilizes learned representations from the pre-trained AntiBERTy language model to directly predict 3D atomic coordinates (Fig. 1). Structures from IgFold are accompanied by a per-residue accuracy estimate, which provides insights into the quality of the prediction.

### Embeddings from pre-trained model encode structural features

The limited number of experimentally determined antibody structures (thousands[31]) presents a difficulty in training an effective antibody structure predictor. In the absence of structural data, self-supervised language models provide a powerful framework for extracting patterns from the significantly greater number (billions[32]) of natural antibody sequences identified by immune repertoire sequencing studies. For this work, we used AntiBERTy[22], a transformer language model pre-trained on 558 million natural antibody sequences, to generate embeddings for structure prediction. Similar to the role played by alignments of evolutionarily related sequences for general protein structure prediction[33], embeddings from AntiBERTy act as a contextual representation that places individual sequences within the broader antibody space.

Prior work has demonstrated that protein language models can learn structural features from sequence pre-training alone[18,34]. To investigate whether sequence embeddings from AntiBERTy contained nascent structural features, we generated embeddings for the set of 3467 paired antibody sequences with experimentally determined structures in the PDB. For each sequence, we extracted the portions of the embedding corresponding to the six CDR loops and averaged to obtain fixed-sized CDR loop representations (one per loop). We then collected the embeddings for each CDR loop across all sequences and visualized using two-dimensional t-SNE (Supplementary Fig. 1). To determine whether the CDR loop representations encoded structural features, we labeled each point according to its canonical structural cluster. For CDR H3, which lacks canonical clusters, we instead labeled by loop length. For the five CDR loops that adopt canonical folds, we observed some organization within the embedded space, particularly for CDR1 loops. For the CDR H3 loop, we found that the embedding space did not separate into natural clusters, but was rather organized roughly in accordance with loop length. These results suggest that AntiBERTy has learned some distinguishing structural features of CDR loops through sequence pre-training alone.

### Coordinate prediction from sequence embeddings

To predict 3D atomic coordinates from sequence embeddings, we adopt a graphical representation of antibody structure, with each

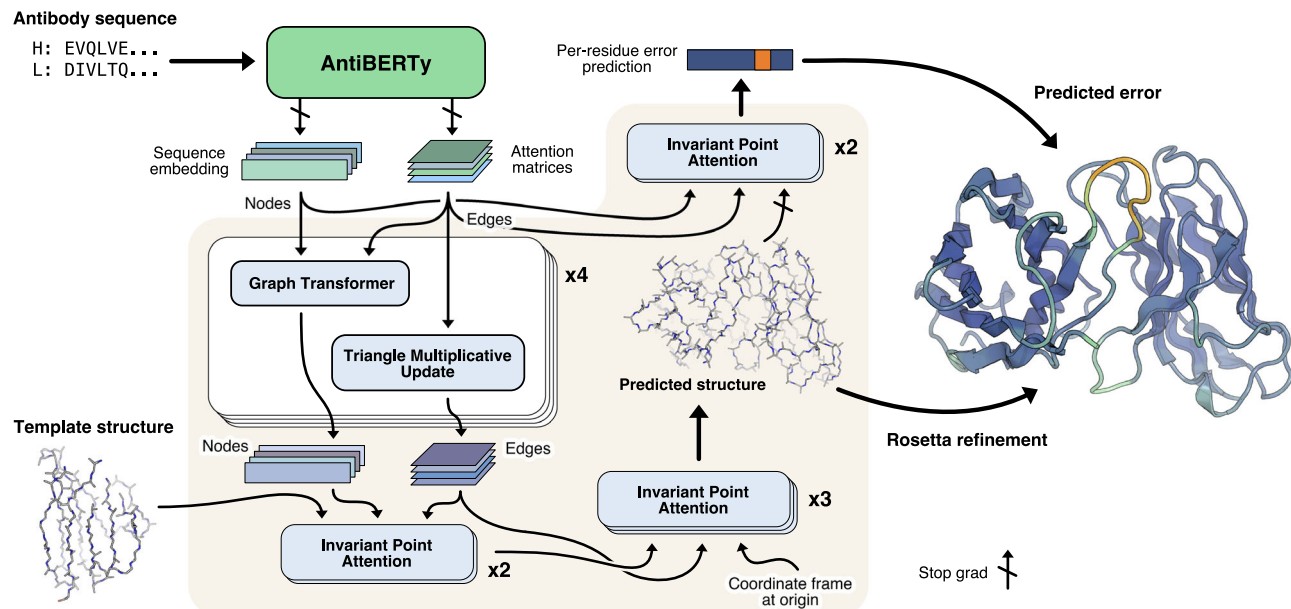

**Fig. 1 | Diagram of method for end-to-end prediction of antibody structures.** Antibody sequences are converted into contextual embeddings using AntiBERTy, a pre-trained language model. From these representations, IgFold uses a series of transformer layers to directly predict atomic coordinates for the protein backbone atoms. For each residue, IgFold also provides an estimation of prediction quality. Refinement of predictions and addition of side chains is performed by Rosetta.

residue as a node and information passing between all pairs of residues (Fig. 1). The nodes are initialized using the final hidden layer embeddings from AntiBERTy. To initialize the edges, we collect the full set of inter-residue attention matrices from each layer of AntiBERTy. These attention matrices are a useful source of edge information as they encode the residue-residue information pathways learned by the pre-trained model. For paired antibodies, we concatenate the sequence embeddings from each chain and initialize inter-chain edges to zero. We do not explicitly provide a chain break delimiter, as the pre-trained language model already includes a positional embedding for each sequence. The structure prediction model begins with a series of four graph transformer[35] layers interleaved with edge updates via the triangle multiplicative layer proposed for AlphaFold[10].

Following the initial graph transformer layers, we incorporate structural template information into the nascent representation using invariant point attention (IPA)[10]. In contrast to the application of IPA for the AlphaFold structure module, we fix the template coordinates and use IPA as a form of structure-aware self-attention. This enables the model to incorporate the local structural environment into the sequence representation directly from the 3D coordinates, rather than switching to an inter-residue representation (e.g., distance or contact matrices). We use two IPA layers to incorporate template information. Rather than search for structural templates for training, we generate template-like structures by corruption of the true label structures. Specifically, for 50% of training examples, we randomly select one to six consecutive segments of twenty residues and move the atomic coordinates to the origin. The remaining residues are provided to the model as a template. The deleted segments of residues are hidden from the IPA attention, so that the model only incorporates structural information from residues with meaningful coordinates.

Finally, we use another set of IPA layers to predict the final 3D antibody structure. Here, we employ a strategy similar to the AlphaFold structure module[10] and train a series of three IPA layers to translate and rotate each residue from an initialized position at the origin to the final predicted position. We depart slightly from the AlphaFold implementation and learn separate weights for each IPA layer, as well as allow gradient propagation through the rotations. To train the model for structure prediction, we minimize the mean-squared error between the predicted coordinates and the experimental structure after Kabsch alignment. In practice, we observe that the first IPA layer is sufficient to learn the global arrangement of residues (albeit in a compact form), while the second and third layers function to produce the properly scaled structure with correct bond lengths and angles (Supplementary Fig. 2).

## Per-residue error prediction

Simultaneously with structure prediction training, we additionally train the model to estimate the error in its own predictions. For error estimation, we use two IPA layers that operate similarly to the template incorporation layers (i.e., without coordinate updates). The error estimation layers take as input the final predicted structure, as well as a separate set of node and edge features derived from the initial AntiBERTy features. We stop gradient propagation through the error estimation layers into the predicted structure to prevent the model from optimizing for accurately estimated, but highly erroneous structures. For each residue, the error estimation layers are trained to predict the deviation of the $N$, $C_\alpha$, $C$, and $C_\beta$ atoms from the experimental structure after a Kabsch alignment of the beta barrel residues. We use a different alignment for error estimation than structure prediction to more closely mirror the conventional antibody modeling evaluation metrics. The model is trained to minimize the L1 norm of the predicted $C_\alpha$ deviation minus the true deviation.

## Structure dataset augmentation with AlphaFold

We sought to train the model on as many immunoglobulin structures as possible. From the Structural Antibody Databae (SAbDab)[31], we obtained 4275 structures consisting of paired antibodies and single-chain nanobodies. Given the remarkable success of AlphaFold for modeling both protein monomers and complexes, we additionally explored the use of data augmentation to produce structures for training. To produce a diverse set of structures for data augmentation, we clustered[36] the paired and unpaired partitions of the Observed Antibody Space[32] at 40% and 70% sequence identity, respectively. This clustering resulted in 16,141 paired sequences and 26,971 unpaired sequences. Because AlphaFold-Multimer[13] was not yet released, all predictions were performed with the original AlphaFold model[10]. For the paired sequences, we modified the model inputs to enable complex modeling by inserting a gap in the positional embeddings (i.e., AlphaFold-Gap[12,13]). For the unpaired sequences, we discarded the predicted structures with average pLDDT (AlphaFold error estimate) <85, leaving 22,132 structures. These low-confidence structures typically correponded to sequences with missing residues at the N-terminus. During training, we sample randomly from the three datasets with examples weighted inversely to the size of their respective datasets, such that roughly one third of total training examples come from each dataset.

## Antibody structure prediction benchmark

To evaluate the performance of IgFold against recent methods for antibody structure prediction, we assembled a non-redundant set of antibody structures deposited after compiling our training dataset. We chose to compare performance on a temporally separated benchmark to ensure that none of the methods evaluated had access to any of the structures during training. In total, our benchmark contains 197 paired antibodies and 71 nanobodies.

## Predicted structures are high quality before refinement

As an end-to-end model, IgFold directly predicts structural coordinates as its output. However, these immediate structure predictions are not guaranteed to satisfy realistic molecular geometries. In addition to incorporating missing atomic details (e.g., side chains), refinement with Rosetta[37] corrects any such abnormalities. To better understand the impact of this refinement step, we compared the directly predicted structures for each target in the benhmark to their refined counterparts. In general, we observed very little change in the structures (Supplementary Fig. 3), with an average RMSD <0.5 Å before and after refinement. The exception to this trend is abnormally long CDR loops, particularly CDR H3. We compared the pre- and post-refinement structures for benchmark targets with three of the longest CDR H3 loops to those with shorter loops and found that the longer loops frequently contained unrealistic bond lengths and backbone torsion angles (Supplementary Fig. 4). Similar issues have been observed in recent previous work[15], indicating that directly predicting atomically correct long CDR loops remains a challenge.

## Accurate antibody structures in a fraction of the time

We compared the performance of IgFold against a mixture of grafting and deep learning methods for antibody structure prediction. Although previous work has demonstrated significant improvements by deep learning over grafting-based methods, we continue to benchmark against grafting to track its performance as increasingly many antibody structures become available. For each benchmark target, we predicted structures using RepertoireBuilder[38], DeepAb[14], ABlooper[15], and AlphaFold-Multimer[13]. We opted to benchmark the ColabFold[12] implementation of AlphaFold, rather than the original pipeline from DeepMind, due to its significant runtime acceleration and similar accuracy. Of these methods, RepertoireBuilder utilizes a grafting-based algorithm for structure prediction and the remaining

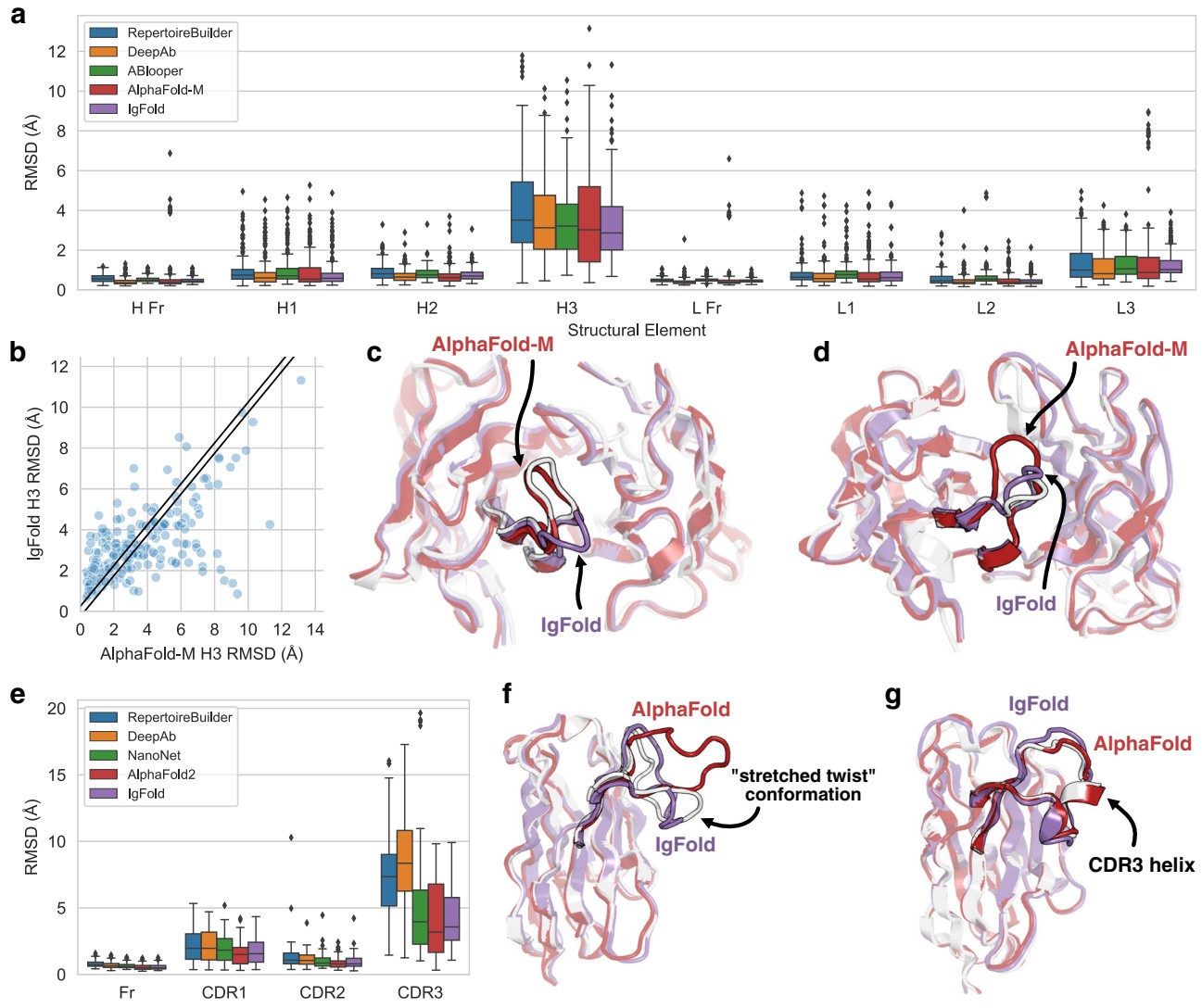

**Fig. 2 | Comparison of methods for antibody structure prediction.** All root-mean-squared-deviation (RMSD) values calculated over backbone heavy atoms after alignment of the respective framework residues. Box plots have center at median, bounds indicating interquartile range (IQR), whisker length of $1.5 \times$ IQR, and poitns outside of $1.5 \times$ IQR range shown as outliers. Source data are provided as a Source Data file. **a** Benchmark performance of RepertoireBuilder, DeepAb, ABlooper, AlphaFold-Multimer, and IgFold for paired antibody structure prediction ($n = 197$ structure predictions). **b** Per-target comparison of CDR H3 loop structure prediction for IgFold and AlphaFold-Multimer, with each point representing the $RMSD_{H3}$ for both methods on a single benchmark target. **c** Comparison of predicted CDR H3 loop structures for target 7N3G ($L_{H3} = 10$ residues) for IgFold ($RMSD_{H3} = 4.69$ Å) and AlphaFold-Multimer ($RMSD_{H3} = 0.98$ Å). **d** Comparison of predicted CDR H3 loop structures for target 7RNJ ($L_{H3} = 9$ residues) for IgFold ($RMSD_{H3} = 1.18$ Å) and AlphaFold-Multimer ($RMSD_{H3} = 3.46$ Å). **e** Benchmark performance of RepertoireBuilder, DeepAb, AlphaFold, and IgFold for nanobody structure prediction ($n = 71$ structure predictions). **f** Comparison of predicted CDR H3 loop structures for target 7AQZ ($L_{CDR3} = 15$ residues) for IgFold ($RMSD_{CDR3} = 2.87$ Å) and AlphaFold ($RMSD_{CDR3} = 7.08$ Å). **g** Comparison of predicted CDR H3 loop structures for target 7ARO ($L_{CDR3} = 17$ residues) for IgFold ($RMSD_{CDR3} = 2.34$ Å) and AlphaFold ($RMSD_{CDR3} = 0.84$ Å).

use some form of deep learning. DeepAb and ABlooper are both trained specifically for paired antibody structure prediction, and have previously reported comparable performance. AlphaFold-Multimer has demonstrated state-of-the-art performance for protein complex prediction—however, performance on antibody structures specifically remains to be evaluated.

The performance of each method was assessed by measuring the backbone heavy-atom (N, $C_\alpha$, C, O) RMSD between the predicted and experimentally determined structures for the framework residues and each CDR loop. All RMSD values are measured after alignment of the framework residues. In general, we observed state-of-the-art performance for all of the deep learning methods while grafting performance continued to lag behind (Fig. 2a, Table 1). On average, all of the antibody-specific methods predicted both the heavy and light chain

framework structures with high accuracy (0.43–0.53 Å and 0.41–0.51 Å, respectively). AlphaFold-Multimer typically performed well on framework residues, except for a set of fourteen predictions where the model predicted C-terminal strand swaps between the heavy and light chains (Supplementary Fig. 5). For the CDR1 and CDR2 loops, all methods produced sub-angstrom predictions on average. The largest improvement in prediction accuracy by deep learning methods is observed for the CDR3 loops.

We also considered the predicted orientation between the heavy and light chains, which is an important determinant of the overall binding surface[8,9]. Accuracy of the inter-chain orientation was evaluated by measuring the deviation from native of the inter-chain packing angle, inter-domain distance, heavy-opening angle, and light-opening angle. Each of these orienational coordinates are rescaled by

**Table 1 | Accuracy of predicted antibody Fv structures**

| Method | OCD | H Fr (Å) | H1 (Å) | H2(Å) | H3 (Å) | L Fr (Å) | L1 (Å) | L2(Å) | L3 (Å) |
|---|---|---|---|---|---|---|---|---|---|
| RepertoireBuilder | 5.09 | 0.59 | 1.00 | 0.90 | 4.15 | 0.49 | 0.81 | 0.57 | 1.32 |
| DeepAb | 3.60 | 0.43 | 0.86 | 0.72 | 3.57 | 0.41 | 0.75 | 0.48 | 1.16 |
| ABlooper | 4.42 | 0.53 | 0.98 | 0.83 | 3.54 | 0.51 | 0.92 | 0.67 | 1.32 |
| AlphaFold-Multimer | 4.18 | 0.69 | 0.95 | 0.74 | 3.56 | 0.66 | 0.84 | 0.51 | 1.59 |
| IgFold | 3.82 | 0.48 | 0.85 | 0.76 | 3.27 | 0.46 | 0.76 | 0.46 | 1.30 |

dividing by their respective standard deviations (calculated over the set of experimentally determined antibody structures) and summed to obtain an orientational coordinate distance (OCD)[9]. We found that in general deep learning methods produced $F_V$ structures with OCD values near four, indicating that the predicted structures are typically within about one standard deviation of the native structures for each of the components of OCD.

Given the comparable aggregate performance of the deep learning methods, we further investigated the similarity between the structures predicted by each method. For each pair of methods, we measured the RMSD of framework and CDR loop residues, as well as the OCD, between the predicted structures for each benchmark target (Supplementary Fig. 9). We additionally plotted the distribution of structural similarities between IgFold and the alternative methods (Supplementary Fig. 10). We found that the framework structures (and their relative orientations) predicted by IgFold resembled those of DeepAb and ABlooper, but were less similar to those of RepertoireBuilder and AlphaFold-Multimer. The similarity between IgFold and ABlooper is expected, given that ABlooper predictions were based on IgFold-predicted framework structures. We also observed that the heavy chain CDR loops from IgFold, DeepAb, and ABlooper were quite similar on average. We observe further similarity on light chain CDRs between IgFold and DeepAb. These agreements likely extend from training on similar, antibody-focused datasets.

**Deep learning methods converge on CDR H3 accuracy**

The average prediction accuracy for the highly variable, conformationally diverse CDR H3 loop was relatively consistent among the four deep learning methods evaluated (Table 1), though IgFold performed the best on average. Given this convergence in performance, we again considered the similarity between the CDR H3 loop structures predicted by each method. IgFold, DeepAb, and ABlooper produced the most similar CDR H3 loops, with an average RMSD of 2.01–2.34 Å between predicted structures for the three methods . This may be reflective of the similar training datasets used for the methods, which were limited to antibody structures. AlphaFold-Multimer, by contrast, predicted the most distinct CDR H3 loops, with an average RMSD 3.10–3.57 Å to the other deep learning methods.

The dissimilarity of predictions between IgFold and AlphaFold-Multimer is surprising, given the extensive use of AlphaFold-predicted structures for training IgFold. When we compared the per-target accuracy of IgFold and AlphaFold-Multimer, we found many cases where one method predicted the CDR H3 loop accurately while the other failed (Fig. 2b). Indeed, ~20% of CDR H3 loops predicted by the two methods were >4 Å RMSD apart, meaning the methods often predict distinct conformations. To illustrate the structural implications of these differences in predictions, we highlight two targets from the benchmark where IgFold and AlphaFold-Multimer diverge. In one such case (target 7N3G[39], Fig. 2c), AlphaFold-Multimer effectively predicts the CDR H3 loop structure ($RMSD_{H3} = 0.98$ Å) while IgFold predicts a distinct, and incorrect, conformation ($RMSD_{H3} = 4.69$ Å). However, for another example (target 7RNJ[40], Fig. 2d), IgFold more accurately predicts the CDR H3 loop structure ($RMSD_{H3} = 1.18$ Å) while AlphaFold-Multimer predicts an alternative conformation ($RMSD_{H3} = 3.46$ Å).

**Table 2 | Accuracy of predicted nanobody structures**

| Method | Fr (Å) | CDR1 (Å) | CDR2 (Å) | CDR3 (Å) |
|---|---|---|---|---|
| RepertoireBuilder | 0.80 | 2.12 | 1.37 | 7.54 |
| DeepAb | 0.72 | 2.14 | 1.14 | 8.52 |
| NanoNet | 0.66 | 1.94 | 1.05 | 5.43 |
| AlphaFold | 0.57 | 1.61 | 0.88 | 4.00 |
| IgFold | 0.58 | 1.73 | 0.98 | 4.25 |

**Fast nanobody structure prediction remains a challenge**

Single domain antibodies, or nanobodies, are an increasingly popular format for therapeutic development[41]. Structurally, nanobodies share many similarities with paired antibodies, but with the notable lack of a second immunoglobulin chain. This, along with increased nanobody CDR3 loop length, makes accessible a wide range of CDR3 loop conformations not observed for paired antibodies[42]. We compared the performance of IgFold for nanobody structure prediction to RepertoireBuilder[38], DeepAb[14], NanoNet[17], and AlphaFold[10] (Fig. 2e, Table 2). We omitted ABlooper from the comparison as it predicts only paired antibody structures.

As with paired antibodies, all methods evaluated produced highly accurate predictions for the framework residues, with the average RMSD ranging from 0.57 Å to 0.80 Å. No method achieves sub-angstrom accuracy on average for CDR1 loops, though AlphaFold and IgFold achieve the best performance. For CDR2 loops, we observe a substantial improvement by IgFold and the other deep learning methods over RepertoireBuilder, with AlphaFold achieving the highest accuracy on average. For the CDR3 loop, RepertoireBuilder prediction quality is highly variable (average $RMSD_{CDR3}$ of 7.54 Å), reflective of the increased difficulty of identifying suitable template structures for the long, conformationally diverse loops. DeepAb achieves the worst performance for CDR3 loops, with an average $RMSD_{CDR3}$ of 8.52 Å, probably because its training dataset was limited to paired antibodies[14], and thus the model has never observed the full range of conformations accessible to nanobody CDR3 loops. NanoNet, trained specifically for nanobody structure prediction, outperforms DeepAb (average $RMSD_{CDR3}$ of 5.43 Å). AlphaFold displays the best performance for CDR3 loops, with an average $RMSD_{CDR3}$ of 4.00 Å, consistent with its high accuracy on general protein sequences. IgFold CDR3 predictions tend to be slightly less accurate than those of AlphaFold (average $RMSD_{CDR3}$ of 4.25 Å), but are significantly faster to produce (15 s for IgFold, versus 6 min for the ColabFold implementation of AlphaFold).

To better understand the distinctions between IgFold- and AlphaFold-predicted nanobody structures, we highlight two examples from the benchmark. First, we compared the structures predicted by both methods for the benchmark target 7AQZ[43] (Fig. 2f). This nanobody features a 15-residue CDR3 loop that adopts the "stretched-twist" conformation[42], in which the CDR3 loop bends to contact the framework residues that would otherwise be obstructed by a light chain in a paired antibody. IgFold correctly predicts this nanobody-specific loop conformation ($RMSD_{CDR3} = 2.87$ Å), while AlphaFold predicts an extended CDR3 conformation ($RMSD_{CDR3} = 7.08$ Å). Indeed, there are

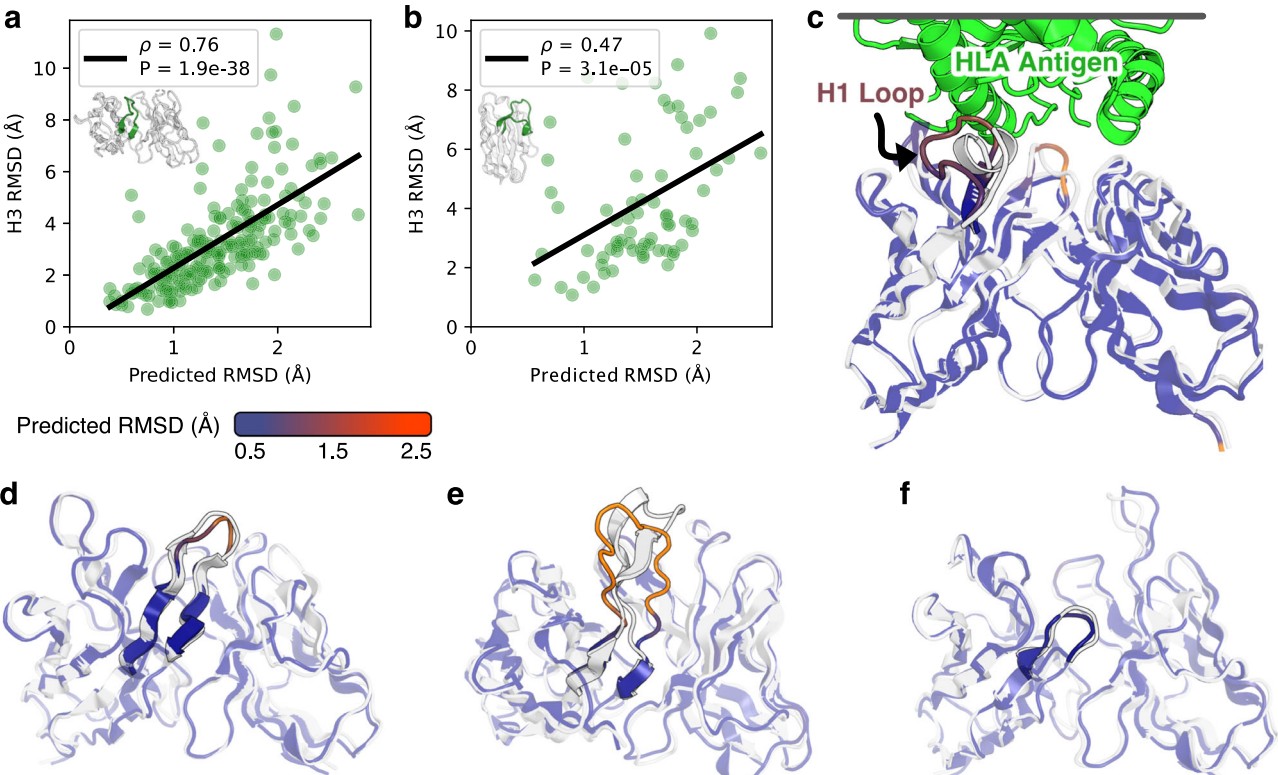

**Fig. 3 | Error estimation for predicted antibody structures.** Reported Spearman correlation coefficients ($\rho$) are between predicted and calculated RMSD values with associated *p* values calculated according to a two-sided *t*-test. Source data are provided as a Source Data file. **a** Comparison of CDR H3 loop RMSD to predicted error for paired antibody structure benchmark. Gray space represents cumulative average RMSD of predicted CDR H3 loops from native structure. **b** Comparison of CDR3 loop RMSD to predicted error for nanobody structure benchmark. Gray space represents cumulative average RMSD of predicted CDR3 loops from native structure. **c** Predicted structure and error estimation for anti-HLA antibody with a randomized CDR H1 loop. **d** Predicted structure and error estimation for benchmark target 7O4Y ($L_{H3}$ = 12 residues). **e** Predicted structure and error estimation for benchmark target 7RKS ($L_{H3}$ = 18 residues). **f** Predicted structure and error estimation for benchmark target 7O33 ($L_{H3}$ = 3 residues).

other cases where either IgFold or AlphaFold correctly predicts the CDR3 loop conformation while the other fails (see off-diagonal points in Supplementary Fig. 8G). In the majority of such cases, AlphaFold predicts the correct conformation, yielding the lower average CDR3 RMSD. In a second example, we compared the structures predicted by both methods for the benchmark target 7ARO[43] (Fig. 2g). This nanobody has a long 17-residue CDR3 loop with a short helical region. Although both methods correctly predict the loop conformation, IgFold fails to predict the helical secondary structure, resulting in a less accurate prediction ($RMSD_{CDR3}$ = 2.34 Å) than that of AlphaFold ($RMSD_{CDR3}$ = 0.84 Å). Such structured loops highlight a key strength of AlphaFold, which was trained on a large dataset of general proteins and has thus encountered a broad variety of structral arrangements, over IgFold, which has observed relatively few such structures within its training dataset.

**Error predictions identify inaccurate CDR loops**

Although antibody structure prediction methods continue to improve, accurate prediction of abnormal CDR loops (particularly long CDR H3 loops) remains inconsistent[6,14,15]. Determining whether a given structural prediction is reliable is critical for effective incorporation of antibody structure prediction into workflows. During training, we task IgFold with predicting the deviation of each residue's $C_\alpha$ atom from the native (under alignment of the beta barrel residues). We then use this predicted deviation as a per-residue error estimate to assess expected accuracy of different structural regions.

To assess the utility of IgFold's error predictions for identifying inaccurate CDR loops, we compared the average predicted error for each CDR loop to the RMSD between the predicted loop and the native

structure for the paired $F_V$ and nanobody benchmarks. We observed significant correlations between the predicted error and the loop RMSDs from native for all the paired $F_V$ CDR loops (Supplementary Fig. 11). For CDR H2 and CDR L2 loops, the correlations between predicted and measured RMSD were notably weaker. However, given the relatively high accuracy of predictions for these loops, there was little error to detect. For nanobodies, we observed significant correlations between the predicted error and RMSD for all the CDR loops (Supplementary Fig. 12). Interestingly, for all loops the model tended to predict lower RMSD than was measured. This may be a result of the imbalance between the smaller number of residues with higher RMSD (CDR loops) and the greater number with lower RMSD (framework residues). In the future, this miscalibration may be solved by using a weighted loss function that penalizes larger errors more heavily. However, the model's ability to effectively rank the accuracy of different CDR loops is still useful for identifying potentially inaccurate predictions.

For the challenging-to-predict, conformationally diverse CDR3 loops, we observed significant correlations for both paired antibody H3 loops (Fig. 3a, $\rho$ = 0.76) and nanobody CDR3 loops (Fig. 3b, $\rho$ = 0.47). To illustrate the utility of error estimation for judging CDR H3 loop predictions, we highlight three examples from the benchmark. The first is the benchmark target 7O4Y[44], a human anti-CD22 antibody with a 12-residue CDR H3 loop. For 7O4Y, IgFold accurately predicts the extended beta sheet structure of the CDR H3 loop ($RMSD_{H3}$ = 1.64 Å), and estimates a correspondingly lower RMSD (Fig. 3d). The second target is 7RKS[45], a human anti-SARS-CoV-2-receptor-binding-domain antibody with a 18-residue CDR H3 loop. IgFold struggles to predict the structured beta sheet within this long

H3 loop, instead predicting a broad ununstructured conformation ($RMSD_{H3}$ = 6.33 Å). Appropriately, the error estimation for the CDR H3 loop of 7RKS is much higher (Fig. 3e). The third example is 7O33[46], a mouse anti-PAS (proine/alanine-rich sequence) antibody with a 3-residue CDR H3 loop. Again, IgFold accurately predicts the structure of this short loop ($RMSD_{H3}$ = 1.49 Å) and provides a correspondingly low error estimate (Fig. 3f).

Antibody engineering campaigns often deviate significantly from the space of natural antibody sequences[47]. Predicting structures for such heavily engineered sequences is challenging, particularly for models trained primarily on natural antibody structural data (such as IgFold). To investigate whether IgFold's error estimations can identify likely mistakes in such sequences, we predicted the structure of an anti-HLA (human leukocyte antigen) antibody with a sequence randomized CDR H1 loop[48] (Fig. 3c). As expected, there is significant error in the predicted CDR H1 loop structure. However, the erroneous structure is accompanied by a high error estimate, revealing that the predicted conformation is likely to be incorrect. This suggests that the RMSD predictions from IgFold are sensitive to unnatural antibody sequences and should be informative for a broad range of antibody structure predictions.

## Template data is successfully incorporated into predictions

For many antibody engineering workflows, partial structural information is available for the antibody of interest. For example, crystal structures may be available for the parent antibody upon which new CDR loops were designed. Incorporating such information into structure predictions is useful for improving the quality of structure models. We simulated IgFold's behavior in this scenario by predicting structures for the paired antibody and nanobody benchmark targets while providing the coordinates of all non-H3 residues as templates. In general, we found that IgFold was able to incorporate the template data into its predictions, with the average RMSD for all templated CDR loops being significantly reduced (IgFold[Fv-H3]: Fig. 4a, IgFold[Fv-CDR3]: Fig. 4c). Although these results are not surprising, they show-case a key functionality lacking in prior antibody-specific methods[14,15,17].

Having demonstrated successful incorporation of structural data into predictions using templates, we next investigated the impact on accuracy of the untemplated CDR H3 loop predictions. For the majority of targets, we found little change in the accuracy of CDR H3 loop structures with the addition of non-H3 template information (Fig. 4b). For nanobodies, we observe more cases with substantial improvement to CDR3 loop predictions given template data (Fig. 4d).

We additionally experimented with providing the entire crystal structure to IgFold as template information. In this scenario, IgFold successfully incorporates the structural information of all CDR loops (including H3) into its predictions (IgFold[Fv]: Fig. 4a, c). Interestingly, the model's incorporation of non-CDR3 templated regions also improves when the full structural context is provided, indicating that the model is not simply recapitulating template structures, but combining their content with its predictions. Although this approach is of little practical value for structure prediction (as the correct structure is already known) it may be a useful approach for instilling structural information into pre-trained embeddings, which are valuable for other antibody learning tasks.

## Minimal refinement yields faster predictions

Although the performance of the deep learning methods for antibody structure prediction is largely comparable, the speed of prediction is not. Grafting-based methods, such as RepertoireBuilder, tend to be much faster than deep learning methods (if a suitable template can be found). However, as reported above, this speed is obtained at the expense of accuracy. Recent deep learning methods for antibody structure prediction, including DeepAb, ABlooper, and NanoNet, have

claimed faster prediction of antibody structures as compared to general methods like AlphaFold. For our benchmark, all deep learning methods were run on identical hardware (12-core CPU with one A100 GPU), allowing us to directly compare their runtimes. All computed runtimes are measured from sequence to full-atom structure, using the recommended full-atom refinement protocols for each method. We could not evaluate the runtimes of RepertoireBuilder as no code has been published. The results of this comparison are summarized in Fig. 4e,f.

For paired antibodies, we find that IgFold is significantly faster than any other method tested. On average, IgFold takes 23 s to predict a full-atom structure from sequence. The next fastest method, ABlooper, averages nearly 3 min (174 s) for full-atom structure prediction. Although ABlooper rapidly predicts coordinates in an end-to-end fashion, the outputs require expensive refinement in OpenMM to correct for geometric abnormalities and add side chains. The ColabFold[12] implementation of AlphaFold-Multimer evaluated here averages just over 7 min (435 s) for full-atom structure prediction. This is considerably faster than the original implementation of AlphaFold-Multimer, which required an expensive MSA search and repeated model compilation for every prediction. Finally, the slowest method for paired antibody structure prediction was DeepAb, which averaged over 12 min (750 s). DeepAb is considerably slower by design, as it requires minimization of predicted inter-residue potentials in Rosetta. We also investigated the impact of sequence length on prediction times. In general, the runtimes of all methods increased with sequence length (Supplementary Fig. 13A). DeepAb and ABlooper were the most sensitive to sequence length, with AlphaFold-Multimer and IgFold scaling more favorably.

For nanobodies, we again find that IgFold outpaces alternative methods for full-atom structure prediction, requiring an average of 15 s. NanoNet was similarly fast, averaging 15 s for full-atom structure prediction. Similar to ABlooper for paired antibodies, NanoNet outputs require expensive refinement to correct for unrealistic backbone geometries and add side chains. DeepAb was able to predict nanobody strucutres in just under 4 min (224 s) on average. Finally, the slowest method for nanobody structure prediction was AlphaFold, which averaged nearly 6 min (345 s). As with paired antibodies, we also investigated the impact of sequence length on prediction times. In general, the runtimes of all methods increased with sequence length (Supplementary Fig. 13B). Although NanoNet had several outlier cases that required significant refinement, the prediction times for a majority of targets increased with sequence length. We also note that for methods capable of predicting both nanobody and paired antibody structures, runtimes tend to roughly double in the paired setting (scaling linearly with total length), as expected.

## Large-scale prediction of paired antibody structures

The primary advantage of IgFold over other highly accurate methods like AlphaFold is its speed at predicting antibody structures. This speed enables large-scale prediction of antibody structures on modest compute resources. Prior work exploring large-scale predictions of antibody structures have provided insight into the structural commonalities across individuals, and provide evidence of a public structural repertoire[49]. Further, comparison on the basis of structure (rather than sequence alone) has enabled discovery of convergent binders that diverge significantly in sequence[50]. To demonstrate the utility of IgFold's speed for such analyses, we predicted structures for two non-redundant sets of paired antibodies. The first set consists of 104,994 paired antibody sequences (clustered at 95% sequence identity) from the OAS database[32]. These sequences are made up of 35,731 human, 16,356 mouse, and 52,907 rat antibodies. The second set contains 1,340,180 unique paired human antibody sequences from the immune repertoires of four unrelated individuals[51]. These sequences span the affinity maturation spectrum, consisting of both naive and memory

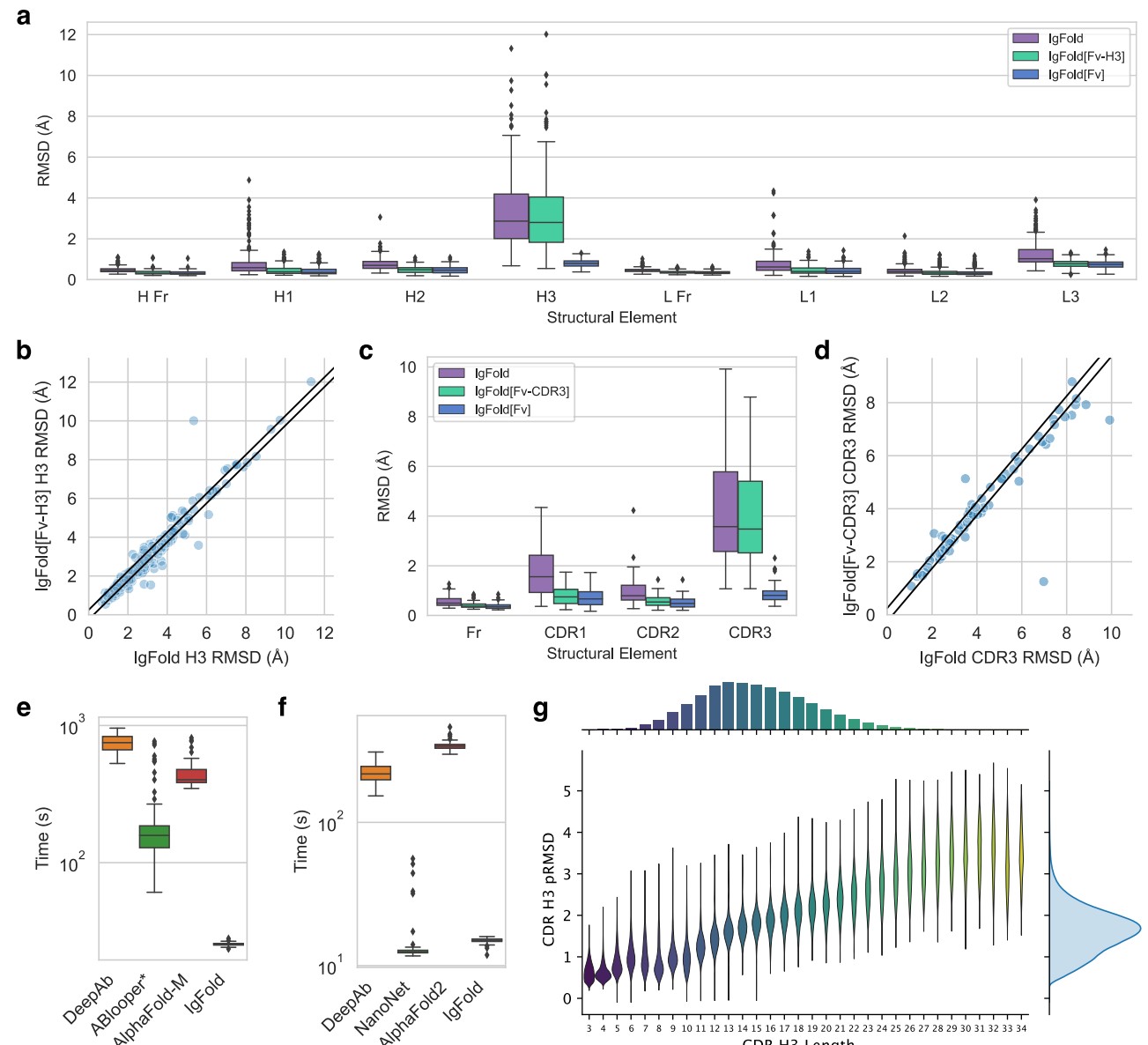

**Fig. 4 | Utility of IgFold for antibody structure prediction.** Box plots have center at median, bounds indicating interquartile range (IQR), whisker length of 1.5 × IQR, and poitns outside of 1.5 × IQR range shown as outliers. Source data are provided as a Source Data file. **a** Paired antibody structure prediction benchmark results ($n$ = 197 structure predictions) for IgFold without templates, IgFold given the $F_V$ structure without the CDR H3 loop (IgFold[Fv-H3]), and IgFold given the complete Fv structure (IgFold[Fv]). **b** Per-target comparison of CDR H3 loop structure prediction for IgFold and IgFold[Fv-H3], with each point representing the RMSD$_{H3}$ for both methods on a single benchmark target. **c** Nanobody structure prediction benchmark results ($n$ = 71 structure predictions) for IgFold without templates, IgFold given the $F_V$ structure without the CDR3 loop (IgFold[Fv-CDR3]), and IgFold given the complete Fv structure (IgFold[Fv]). **d** Per-target comparison of CDR3 loop structure prediction for IgFold and IgFold[Fv-CDR], with each point representing the RMSD$_{CDR3}$ for both methods on a single benchmark target. **e** Runtime comparison of evaluated methods on the paired antibody structure prediction benchmark ($n$ = 197 structure predictions). ABlooper runtimes are calculated given an IgFold-predicted framework, and thus represent an underestimation of actual runtime (**f**) Runtime comparison of evaluated methods on the nanobody structure prediction benchmark ($n$ = 71 structure predictions). **g** Distribution of predicted RMSD and CDR H3 loop lengths for 1.3 million predicted human paired antibody structures.

B-cell sequences. The structures are predicted with low estimated RMSD by IgFold, indicating that they are accurate (Supplementary Fig. 14, Supplementary Fig. 15). We highlight the predicted accuracy of the CDR H3 loops for the 1.3 million human antibody sequences in Fig. 4g. The median length and predicted RMSD for this set are 13 residues and 1.95 Å, respectively. We note that the predicted RMSD values tend to be underestimations, and in practice the actual H3 loop RMSDs, were structures to be experimentally determined, would likely be higher. As of October 2022, only 2448 unique paired antibody structures have been determined experimentally[31], and thus our predicted dataset represents an over 500-fold expansion of antibody

structural space. These structures are made available for use in future studies.

## Discussion

Protein structure prediction methods have advanced significantly in recent years, and they are now approaching the accuracy of the experimental structures upon which they are trained[10]. These advances have been enabled in large part by effective exploitation of the structural information present in alignments of evolutionarily related sequences (MSAs). However, constructing a meaningful MSA is time-consuming, contributing significantly to the runtime of

general protein structure prediction models, and making high-throughput prediction of many protein structures computationally prohibitive for many users. In this work, we presented IgFold: a fast, accurate model that specializes in prediction of antibody structures. We demonstrated that IgFold matches the accuracy of the highly accurate AlphaFold-Multimer model[13] for paired antibody structure prediction, and approaches the accuracy of AlphaFold for nanobodies. Though prediction accuracy is comparable, IgFold is significantly faster than AlphaFold, and is able to predict structures in seconds. Further, for many targets IgFold and AlphaFold predict distinct conformations, which should be useful in assembling structural ensembles for applications where flexibility is important. Predicted structures are accompanied by error estimates, which provide critical information on the reliability of structures.

Analyses of immune repertoires have traditionally been limited to sequence data alone[1], as high-throughput antibody structure determination was experimentally prohibitive and prediction methods were too slow or inaccurate. However, incorporation of structural context has proven valuable, particularly for identification of sequence-dissimilar binders to common epitopes[52]. For example, grafting-based methods have been used to identify sequence-diverse but structurally similar antibodies against SARS-CoV-2[50]. The increased accuracy of IgFold, coupled with its speed, will make such methods more effective. In addition, consideration of structural uncertainty via IgFold's error estimation should reduce the rate of false positives when operating on large volumes of sequences. As a demonstration of IgFold's capabilities, we predicted structures for over 1.4 million paired antibody sequences spanning three species. These structures expand on the number of experimentally determined antibody structures by a factor of 500. The majority of these structures are predicted with high confidence, suggesting that they are reliable. Although our analysis of these structures was limited, we are optimistic that this large dataset will be useful for future studies and model development.

Despite considerable improvements by deep learning methods for general protein complex prediction, prediction of antibody-antigen binding remains a challenge. Even the recent AlphaFold-Multimer model, which can accurately predict the interactions of many proteins, is still unable to predict how or whether an antibody will bind to a given antigen[13]. One of the key barriers to training specialized deep learning models for antibody-antigen complex prediction is the limited availability of experimentally determined structures. The large database of predicted antibody structures presented in this work may help reduce this barrier if it can be employed effectively. In the meantime, IgFold will provide immediate benefits to existing antibody-antigen docking methods. For traditional docking methods, the improvements to speed and accuracy by IgFold should be sufficient to make them more effective[53,54]. For newer docking methods that incorporate structural flexibility, the error estimates from IgFold may be useful for directing enhanced sampling[55].

Deep learning methods trained on antibody sequences and structures hold great promise for design of novel therapeutic and diagnostic molecules. Generative models trained on large numbers of natural antibody sequences can produce effective libraries for antibody discovery[29,30]. Self-supervised models have also proven effective for humanization of antibodies[28]. Meanwhile, methods like AlphaFold and RoseTTAFold have been adapted for gradient-based design of novel protein structures and even scaffolding binding loops[56,57]. IgFold will enable similar applications, and will additionally be useful as an oracle to test or score novel antibody designs. Finally, embeddings from IgFold (particularly when injected with structural information from templates) will be useful features for future antibody design tasks.

**Table 3 | IgFold hyperparameters**

| Parameter | Value | Description |
|---|---|---|
| $d_{node}$ | 64 | Node dimension |
| $d_{edge}$ | 64 | Edge dimension |
| $d_{gt-head}$ | 32 | Graph transformer attention head dimension |
| $n_{gt-head}$ | 8 | Graph transformer attention head number |
| $d_{gt-ff-dim}$ | 256 | Graph transformer feedforward transition dimension |
| $n_{gt-layers}$ | 4 | Graph transformer layers |
| $d_{ipa-temp-head-scalar}$ | 16 | Template IPA scalar attention head dimension |
| $d_{ipa-temp-head-point}$ | 4 | Template IPA point attention head dimension |
| $n_{ipa-temp-head}$ | 8 | Template IPA attention head number |
| $d_{ipa-temp-ff-dim}$ | 64 | Template IPA feedforward transition dimension |
| $d_{ipa-temp-ff-layers}$ | 3 | Template IPA feedforward transition layers |
| $n_{ipa-temp-layers}$ | 2 | Template IPA layers |
| $d_{ipa-str-head-scalar}$ | 16 | Structure IPA scalar attention head dimension |
| $d_{ipa-str-head-point}$ | 4 | Structure IPA point attention head dimension |
| $n_{ipa-str-head}$ | 8 | Structure IPA attention head number |
| $d_{ipa-str-ff-dim}$ | 64 | Structure IPA feedforward transition dimension |
| $d_{ipa-str-ff-layers}$ | 3 | Structure IPA feedforward transition layers |
| $n_{ipa-str-layers}$ | 3 | Structure IPA layers |
| $d_{ipa-err-head-scalar}$ | 16 | Error prediction IPA scalar attention head dimension |
| $d_{ipa-err-head-point}$ | 4 | Error prediction IPA point attention head dimension |
| $n_{ipa-err-head}$ | 4 | Error prediction IPA attention head number |
| $d_{ipa-err-ff-dim}$ | 64 | Error prediction IPA feedforward transition dimension |
| $d_{ipa-err-ff-layers}$ | 3 | Error prediction IPA feedforward transition layers |
| $n_{ipa-err-layers}$ | 2 | Error prediction IPA layers |

## Methods

### Generating AntiBERTy embeddings

To generate input features for structure prediction, we use the pre-trained AntiBERTy language model[22]. AntiBERTy is a bidirectional transformer trained by masked language modeling on a set of 558 million antibody sequences from the Observed Antibody Space. For a given sequence, we collect from AntiBERTy the final hidden layer state and the attention matrices for all layers. The hidden state of dimension $L \times 512$ is reduced to dimension $L \times d_{node}$ by a fully connected layer. The attention matrices from all 8 layers of AntiBERTy (with 8 attention heads per layer) are stacked to form an $L \times L \times 64$ tensor. The stacked attention tensor is transformed to dimension $L \times L \times d_{edge}$ by a fully connected layer.

### IgFold model implementation

The architecture and training procedure for IgFold are described below. Full details of the model architecture hyperparameters are detailed in Table 3. In total, IgFold contains 1.6M trainable parameters.

The IgFold model takes as input per-residue embeddings (nodes) and inter-residue attention features (edges). These initial features are processed by a series node updates via graph transformer layers[35] and edge updates via triangular multiplicative operations[10]. Next, template data are incorporated via fixed-coordinate invariant point attention. Finally, the processed nodes and edges are used to predict the antibody backbone structure via invariant point attention. We detail each of these steps in the following subsections. Where possible, we use the same notation as in the original papers.

Residue node embeddings are updated by graph transformer (GT) layers, which extend the powerful transformer architecture to include edge information[35]. Each GT layer takes as input a series of node

embeddings $H^{(l)} = \{h_1, h_2, \ldots, h_L\}$, with $h_i \in \mathbb{R}^{d_{\text{node}}}$, and edges $e_{ij} \in \mathbb{R}^{d_{\text{edge}}}$. We calculate the multi-head attention for each node $i$ to all other nodes $j$ as follows:

$$q_{c,i} = \mathbf{W}_{c,q} h_i \tag{1}$$

$$k_{c,j} = \mathbf{W}_{c,k} h_j \tag{2}$$

$$e_{c,ij} = \mathbf{W}_{c,e} e_{ij} \tag{3}$$

$$\alpha_{c,ij} = \frac{\langle q_{c,i}, k_{c,j} + e_{c,ij} \rangle}{\sum_{u \in L} \langle q_{c,i}, k_{c,u} + e_{c,iu} \rangle} \tag{4}$$

where $\mathbf{W}_{c,q}, \mathbf{W}_{c,k}, \mathbf{W}_{c,e} \in \mathbb{R}^{d_{\text{node}} \times d_{\text{gt-head}}}$ are learnable parameters for the key, query, and edge tranformations for the $c$-th attention head with hidden size $d_{\text{gt-head}}$. In the above, $\langle q, k \rangle = \exp \frac{q^T k}{\sqrt{d}}$ is the exponential of the standard scaled dot product attention operation. Using the calculated attention, we aggregate updates from all nodes $j$ to node $i$ as follows:

$$v_{c,j} = \mathbf{W}_{c,v} h_j \tag{5}$$

$$\hat{h}_i = \|_c^C \left[ \sum_{j \in L} \alpha_{c,ij}(v_{c,j} + e_{c,ij}) \right] \tag{6}$$

where $\mathbf{W}_{c,v} \in \mathbb{R}^{d_{\text{node}} \times d_{\text{gt-head}}}$ is a learnable parameter for the value transformation for the $c$-th attention head. In the above, $\|$ is the concatenation operation over the outputs of the $C$ attention heads. Following the original GT, we use a gated residual connection to combine the updated node embedding with the previous node embedding:

$$\beta_i = \text{sigm}(\mathbf{W}_g[\hat{h}_i; h_i; \hat{h}_i - h_i]) \tag{7}$$

$$h_i^{\text{new}} = (1 - \beta_i) h_i + \beta_i \hat{h}_i \tag{8}$$

where $\mathbf{W}_g \in \mathbb{R}^{3*d_{\text{node}} \times 1}$ is a learnable parameter that controls the strength of the gating function.

Inter-residue edge embeddings are updated using the efficient triangular multiplicative operation proposed for AlphaFold[10]. Following AlphaFold, we first calculate updates using the "outgoing" triangle edges, then the "incoming" triangle edges. We calculate the outgoing edge transformations as follows:

$$a_{ij} = \text{sigm}(\mathbf{W}_{a,g} e_{ij}) \mathbf{W}_{a,v} e_{ij} \tag{9}$$

$$b_{ij} = \text{sigm}(\mathbf{W}_{b,g} e_{ij}) \mathbf{W}_{b,v} e_{ij} \tag{10}$$

where $\mathbf{W}_{a,v}, \mathbf{W}_{b,v} \in \mathbb{R}^{d_{\text{edge}} \times 2*d_{\text{edge}}}$ are learnable parameters for the transformations of the "left" and "right" edges of each triangle, and $\mathbf{W}_{a,g}, \mathbf{W}_{b,g} \in \mathbb{R}^{d_{\text{edge}} \times 2*d_{\text{edge}}}$ are learnable parameters for their respective gating functions. We calculate the outgoing triangle update for edge $ij$ as follows:

$$g_{ij}^{\text{out}} = \text{sigm}(\mathbf{W}_{c,g}^{out} e_{ij}) \tag{11}$$

$$\hat{e}_{ij}^{\text{out}} = g_{ij}^{\text{out}} \odot \mathbf{W}_{c,v}^{\text{out}} \sum_{k \in L}(a_{ik} \odot b_{jk}) \tag{12}$$

$$e_{ij}^{\text{new}} = e_{ij} + \hat{e}_{ij}^{\text{out}} \tag{13}$$

where $\mathbf{W}_{c,v}^{out} \in \mathbb{R}^{2*d_{\text{edge}} \times d_{\text{edge}}}$ and $\mathbf{W}_{c,g}^{\text{out}} \in \mathbb{R}^{d_{\text{edge}} \times d_{\text{edge}}}$ are learnable parameters for the value and gating transformations, respectively, for the outgoing triangle update to edge $e_{ij}$. After applying the outgoing triangle update, we calculate the incoming triangle update similarly as follows:

$$g_{ij}^{\text{in}} = \text{sigm}(\mathbf{W}_{c,g}^{\text{in}} e_{ij}) \tag{14}$$

$$\hat{e}_{ij}^{\text{in}} = g_{ij}^{\text{in}} \odot \mathbf{W}_{c,v}^{\text{in}} \sum_{k \in L}(a_{ki} \odot b_{kj}) \tag{15}$$

$$e_{ij}^{\text{new}} = e_{ij} + \hat{e}_{ij}^{\text{in}} \tag{16}$$

where $\mathbf{W}_{c,v}^{\text{in}} \in \mathbb{R}^{2*d_{\text{edge}} \times d_{\text{edge}}}$ and $\mathbf{W}_{c,g}^{\text{in}} \in \mathbb{R}^{d_{\text{edge}} \times d_{\text{edge}}}$ are learnable parameters for the value and gating transformations, respectively, for the incoming triangle update to edge $e_{ij}$. Note that $a_{ij}$ and $b_{ij}$ are calulated using separate sets of learnable parameters for the outgoing and incoming triangle updates.

To incorporate structural template information into the node embeddings, we adopt the invariant point attention (IPA) algorithm proposed for AlphaFold[10]. Template information is incorporated using a block of two IPA layers, with each containing an attention operation and a three-layer feedforward transition block. For IPA layers, attention between residues is calculated using self-attention from the node embeddings, pairwise bias from the edge embeddings, and projected vectors from the local frames of each residue. Because our objective is to incorporate known structural data into the embedding, frames are not updated between IPA layers. We incorporate partial structure information by masking the attention between residue pairs that do not both have known coordinates. As a result, when no template information is provided, the node embeddings are updated only using the transition layers.

The processed node and edge embeddings are passed to a block of three IPA layers to predict the residue atomic coordinates. We adopt a "residue gas" representation, in which each residue is represented by an independent coordinate frame. The coordinate frame for each residue is defined by four atoms (N, $C_\alpha$, C, and $C_\beta$) placed with ideal bond lengths and angles. We initialize the structure with all residue frames having $C_\alpha$ at the origin and task the model with predicting a series of translations and rotations that assemble the complete structure.

### Training procedure

The model is trained using a combination of structure prediction and error estimation loss terms. The primary structure prediction loss is the mean-squared-error between the predicted residue frame atom coordinates (N, $C_\alpha$, C, and $C_\beta$) and the label coordinates after Kabsch alignment of all atoms. We additionally apply an L1 loss to the inter-atomic distances of the $(i, i+1)$ and $(i, i+2)$ backbone atoms to encourage proper bond lengths and secondary structures. Finally, we use an L1 loss for error prediction, where the label error is calculated as the $C_\alpha$ deviation of each residue after Kabsch alignment of all atoms belonging to beta sheet residues. The total loss is the sum of the structure prediction loss, the inter-atomic distance loss, and the error prediction loss:

$$\text{Loss}(x_{\text{pred}}, x_{\text{label}}) = L_{\text{coords}}(x_{\text{pred}}, x_{\text{label}}) + \text{clamp}(10 \times L_{\text{bonds}}(x_{\text{pred}}), 1)$$
$$+ L_{\text{error}}(x_{\text{pred}}, x_{\text{label}}) \tag{17}$$

where $x_{pred}$ and $x_{label}$ are the predicted and experimentally determined structures, respectively. We scale the bond length loss by a factor of 10 (effectively applying the loss on the nanometer scale) and clamp losses >1. Clamping the bond length loss allows the model to learn global arrangement of residues early in training then improve smaller details (e.g., bond lengths) later in training.

During training we sampled structures evenly between the SAb-Dab dataset[31] and the paired and unpaired synthetic structure datasets. We held out 10% of the SAbDab structures for validation during training. We used the RAdam optimizer[58] with an initial learning rate of $5 \times 10^{-4}$, with learning rate decayed on a cosine annealing schedule. We trained an ensemble of four models with different random seeds. Each model trained for $2 \times 10^6$ steps, with a batch size of one structure. Training took approximately 110 h per model on a single A100 GPU.

### Ensemble structure prediction
To generate a structure prediction for a given sequence, we first make predictions with each of the four ensemble models. We then use the predicted error to select a single structure from the set of four. Rather than use the average predicted error over all residues, we instead rank the structures by the 90th percentile residue error. Typically, the 90th percentile residue error corresponds to the challenging CDR3 loop. Thus, we effectively select the structure with the lowest risk of significant error in the CDR3 loop.

### Refinement procedure
Predicted structures from the IgFold model undergo two stages of refinement to resolve non-realistic features and add side-chain atoms. First, the backbone structure is optimized in PyTorch using a loss function consisting of idealization terms and an RMSD constraint:

$$\text{Loss}(x_{ref}, x_{pred}) = L_{bond-length}(x_{ref}) + L_{bond-angle}(x_{ref}) \\ + L_{peptide-dihedral}(x_{ref}) + L_{coords}(x_{ref}, x_{pred}) \tag{18}$$

where $x_{ref}$ and $x_{pred}$ are the updated and originally predicted structures, respectively. We optimize bond lengths and planar angles according to the standard values reported by Engh and Huber[59]. The peptide bond dihedral angle is optimized to be in the trans conformation. The coordinate loss term is the same as used in model training, but instead of measuring deviation from an experimentally determined structure, it is constraining the updated structure to stay close to the original model prediction. The refinement is performed using the Adam optimizer[60] with a learning rate of 0.02 for 80 steps. Next, the structure is refined in Rosetta using the standard ref2015 energy function[37]. Rosetta refinement progresses through three stages: (1) full-atom energy minimization, (2) side-chain repacking, (3) full-atom energy minimization. Each minimization stage is performed for 100 steps with constraints to the starting coordinates.

### Benchmark datasets
To evaluate the performance of IgFold and other antibody structure prediction methods, we collected a set of high-quality paired and single-chain antibody structures from SAbDab. To ensure none of the deep learning models were trained using structures in the benchmark, we only used structures deposited between July 1, 2021, and September 1, 2022, (after DeepAb, ABlooper, AlphaFold, and IgFold were trained). Structures were filtered at 99% sequence identity. From these structures, we selected those with resolution >3.0 Å. Finally, we removed structures with CDR H3 loops longer than 20 residues (according to Chothia numbering). These steps resulted in 197 paired and 71 single-chain antibody structures for benchmarking methods.

### Benchmarking alternative methods
We compared the performance of IgFold to five alternative methods for antibody structure prediction: RepertoireBuilder, DeepAb, ABlooper, NanoNet, and AlphaFold. RepertoireBuilder structures were predicted using the web server, omitting structures released after July 1, 2021 (benchmark collection date). All of the following methods were run on identical computational hardware, with a 12-core CPU and one A100 GPU. DeepAb structures are generated using the public code repository, with five decoys per sequence as recommended in the publication[14]. ABlooper structures are predicted using the public code repository, with CDR loops built onto frameworks predicted by IgFold. We diverge from the original publication's usage of ABodyBuilder[61] for predicting framework strucutres because the ABodyBuilder web server does not permit omission of enough template structures to perform proper benchmarking (and no code is available). Instead, we used IgFold framework structures because the model did not produce any outliers or failures on these residues. ABlooper predictions were refined using the provided OpenMM[62] pipeline. NanoNet structures were predicted using the public code repository[17], with full-atom refinement processing performed using the provided MODELLER[63] pipeline. AlphaFold (and AlphaFold-Multimer) structures were predicted using the optimized ColabFold repository[12]. The ColabFold pipeline utilizes the model weights trained by DeepMind, but replaces the time-consuming MSA generation step with a faster search via MMseqs2[64]. For both AlphaFold and AlphaFold-Multimer, we made predictions with all five pre-trained models and selected the highest-ranking structure for benchmarking.

### Reporting summary
Further information on research design is available in the Nature Portfolio Reporting Summary linked to this article.

### Data availability
The structures used for training IgFold have been deposited in the Zenodo database under accession code 10.5281/zenodo.7820263 [https://doi.org/10.5281/zenodo.7820263]. The structure prediction data used for benchmarking in this study have been deposited in the Zenodo database under accession code 10.5281/zenodo.7677723 [https://doi.org/10.5281/zenodo.7677723]. Paired antibody structures predicted by IgFold for the 104 thousand OAS sequences and 1.3 million human sequences are available at [https://github.com/Graylab/IgFold]. Experimentally determined structures used for model training and evaluation were accessed from the Protein Data Bank (PDB) [https://www.rcsb.org]. Natural antibody sequences used for data augmentation were accessed from the Observed Antibody Space (OAS) [https://opig.stats.ox.ac.uk/webapps/oas/]. Source data are provided with this paper.

### Code availability
Code and pre-trained models for IgFold are available at [https://github.com/Graylab/IgFold] and have been deposited in the Zenodo database under accession code 10.5281/zenodo.7709609 [https://doi.org/10.5281/zenodo.7709609].

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

## Acknowledgements

We thank Dr. Jeremias Sulam (JHU) and Richard Shuai (UC Berkeley) for helpful discussions throughout this work, Brennan Abanades (Oxford) for assistance with making ABlooper predictions, and Deniz Akpinaroglu (UCSF) for comments on an early version of this paper. This work was supported by National Institutes of Health grants R01-GM078221 (J.A.R., L.-S.C.) and R35-GM141881 (all authors) and AstraZeneca (J.A.R.). Com-putational resources were provided by the Advanced Research Com-puting at Hopkins (ARCH).

## Author contributions

J.A.R. and J.J.G. conceptualized the project. J.A.R., L.-S.C, S.P.M., and J.J.G. contributed to the methodology. J.A.R. developed the software and conducted the investigation. J.J.G. supervised the project. J.A.R. wrote the original paper. J.A.R., L.-S.C, S.P.M., and J.J.G. edited the paper.

## Competing interests

J.J.G. and J.A.R. are inventors on the IgFold technology developed in this study. Johns Hopkins University, J.J.G. and J.A.R. may be entitled to a portion of revenue received on commercial licensing of IgFold. This arrangement has been approved by the Johns Hopkins University in accordance with its competing interests policies. The remaining authors declare no competing interests.
