## [Peer review file · Nature Communications]

REVIEWER COMMENTS

Reviewer #1 (Remarks to the Author):

Summary:

The authors present a novel deep-learning method for fast antibody and nanobody structure prediction with accuracy estimation. Their method can be divided into three main blocks: an antibody language model (described in previous publication), a graph transformer with edge updates and a structure module. Their model produces antibody structures of comparable accuracy to AlphaFold Multimer in a fraction of the time. Their model is also capable of producing nanobody structures of comparable accuracy to those predicted by AlphaFold2.

It is a well written paper, easy to read and with clear figures.

Major comments:

1. In the abstract, the authors make the following statement: “we predicted structures for 105K paired antibody sequences, expanding the observed antibody structural space by over 40-fold.”. Large scale modelling of antibody sequences has been done previously (M. Raybould, 2021). This statement needs to be removed or qualified in some way
2. In the abstract, the authors make the following statement: “Accurate structure prediction on this timescale makes possible avenues of investigation that were previously infeasible”. However, in the paper they show that IGFold has about equal speed and around a 10% increase in accuracy with respect to an already published method ABlooper. This statement needs to be removed or rephrased to say equally fast but improved performance .
3. Why is AlphaFold2 used to generate the augmented structural dataset and AlphaFold Multimer for the benchmark?
4. The template data is successfully incorporated into predictions section is confusing. Is it surprising that given the true values for certain parts of the structure the model will be capable of predicting these accurately?

5. For nanobody structure prediction the authors should compare to Nanonet (T. Cohen, 2021) which claims higher accuracy than AlphaFold2 and greater speed than that claimed in this paper.

6. The authors claim that homology modelling is less accurate for the framework (line 25). However their results show otherwise (AbodyBuilder the homology modelling methods that is compared to is more accurate at predicting the light chain framework than IgFold in the used benchmark)

7. In line 312 the authors claim that IgFold approaches the accuracy of AlphaFold2 on nanobodies. From the results it appears that AlphaFold is still a far more accurate on these proteins.

8. The code is provided under a non-commercial licence.

Minor comments:

1. When showing clusters generated from AntiBERTy embeddings for each CDR, the authors argue that the clustering plots show that the encodings include structural features. However, the claim that the organization observed within the embedding space corresponds to different canonical folds is debatable. Furthermore, for CDR-H3 sequences are being labelled based on CDR length.

2. In line 269, the authors claim that their predicted RMSD is well-calibrated with respect to the real RMSD. Although they show that both these metrics do correlate, in every plot where they compare these values the predicted RMSD consistently predicts lower values than the true RMSD. The authors should at least comment on this.

3. When showing the predicted RMSD for the generated structural OAS dataset, it would be beneficial to split this by CDR, as the whole Fv RMSD is often uninformative.

7. There is a typo in the caption of Figure 4. It should be structure instead of structre.

Reviewer #2 (Remarks to the Author):

[Summary of the contribution] The paper presents IgFold, a deep learning method for antibody structure prediction. IgFold is an end-to-end neural network that consists of a pre-trained language model trained on 558M natural antibody sequences, followed by graph networks that directly predict backbone atom coordinates. The network allows for ingestion of template information and provides uncertainty estimates in addition to the residue locations. IgFold performance was compared with four baseline models. The results show that the accuracy is lower than AlphaFold-Multimer, similar to DeepAb, and better than two non-deep learning baselines. The authors mention that IgFold provides faster predictions than AlphaFold (about 50 times faster) and DeepAb (about 10 times faster) and is comparable in time to the other two baselines.

[Technical contribution] The proposed neural network combines good elements of previously proposed networks such as AlphaFold. Thus, the paper does not have novelty with respect to advancing state of the art in deep learning. However, it is a reasonable application of deep learning to this particular bioinformatics problem. The value of this paper should be found in its potential to advance science.

[Strengths]

- + The main strength of IgFold is that it provides good accuracy and fast predictions. So, it fills a gap between very accurate but slower AlphaFold and less accurate but faster ABodyBuilder. Thus, IgFold can be helpful in specific applications where gains in computation efficiency justify losses in accuracy.
- + Presentation is very good. The paper is well organized, and everything is explained sufficiently well. Analysis of predictions on specific antibodies provides an insight into the performance of IgFold and baselines.
- + Sharing over 100k predictions with the research community is welcome

[Weaknesses and Suggestions]

- Main weakness of the work is the lower accuracy of IgFold compared to AlphaFold. To me, this is the paper's most interesting result: AlphaFold, trained on a large set of structures without an attempt to specialize in antibody structures, is more accurate than IgFold, explicitly trained to do well on antibodies. Such a result is not unheard of in deep learning because foundational models in other domains (e.g., in NLP and computer vision) also often do better in specialized applications than the smaller models made specifically for those specialized applications. [Suggestion] It will be helpful for the authors to point to that similarity with other domains.
- The authors attempt to justify IgFold by comparing prediction speed. Unfortunately, the computational time is mentioned only briefly in lines 172-180 and 217, which does not amount to a thorough analysis.

[Suggestion 1] To make this point stronger, the authors should devote more space to this aspect. It would be good to see the prediction times of each model on each of the test structures and see how the sequence length impact the prediction times. [Suggestion 2] Because low prediction cost is stated as the objective of this paper, the authors should change how they present their work. It would be important to discuss how the authors designed IgFold with that objective in mind. It would be important to provide an ablation study that demonstrates (1) that reducing IgFold architecture comes at the cost of unacceptable deterioration in accuracy and (2) that additional layers do not lead to further improvements in accuracy. [Suggestion 3] It would be important for the authors to provide more detail about why AlphaFold and DeepAb are slower than IgFold. In particular, what are the main reasons the baselines are slower, and would removing or compressing some of their components cause significant loss of accuracy?

- The authors repeatedly imply (lines 199, 227, 236) that IgFold predictions could be helpful despite being less accurate than AlphaFold. The argument seems to be that IgFold predictions improve the diversity of predicted structures, which could be helpful in some downstream applications. This claim is suspect because it is well known from the mixture of experts research that adding inferior predictors to an ensemble leads to lower accuracy. [Suggestion] For a claim of that sort to hold, the authors should show at least one example (hopefully, not cherry-picked) where adding IgFold to AlphaFold predictions is helpful.

- The authors imply that IgFold finds a sweet spot between accuracy and prediction time. However, they do not provide any specific practical example where IgFold would be desirable over more accurate or faster models. [Suggestion] The authors should introduce a cost function that combines accuracy and time to demonstrate it. For example, in what application 10% accuracy loss can be justified by 10 or 100 times lower computational cost?

- On one hand, it is positive that the authors used the structures discovered after July 1, 2021, as test data in the evaluation. On another, the set of 67 paired antibodies and 21 nanobodies is too small to evaluate IgFold properly. [Suggestion] It would be important to create the second, larger test data set using structures prior to July 1, 2021. One suggestion is to start selecting test structures from the training data one by one at random and remove from the training data set all similar (e.g., 70% and higher identity in CDRs) sequences. In this way, the test sequences will be sufficiently different from the remaining training data and would allow proper testing without the fear of biased results.

+ The results shown in Figures 3 and 4 and the accompanying discussion only refer to 4 specific structures. It is not clear why those structures are selected over other test structures. Without justification, a reader can start wondering if those structures are cherry-picked.

Response to reviewers

We thank the reviewers for their comments and critiques, which have undoubtedly resulted in a stronger manuscript. Below we detail the changes made in response to the reviewer's comments. For convenience, the original reviewer comments are included, while our responses (indented) are below. Changes to the manuscript are indicated by **red text**.

Reviewer 1

Summary:

The authors present a novel deep-learning method for fast antibody and nanobody structure prediction with accuracy estimation. Their method can be divided into three main blocks: an antibody language model (described in previous publication), a graph transformer with edge updates and a structure module. Their model produces antibody structures of comparable accuracy to AlphaFold Multimer in a fraction of the time. Their model is also capable of producing nanobody structures of comparable accuracy to those predicted by AlphaFold2.

It is a well written paper, easy to read and with clear figures.

Major comments:

In the abstract, the authors make the following statement: “we predicted structures for 105K paired antibody sequences, expanding the observed antibody structural space by over 40-fold.”. Large scale modelling of antibody sequences has been done previously (M. Raybould, 2021). This statement needs to be removed or qualified in some way

We appreciate the reviewer bringing this prior work to our attention. We have added a discussion of this work to the results section, to highlight the value of large-scale modeling of antibody structures. Additionally, we have now significantly expanded our set of predicted antibody structures to 1.4M unique structures. To our knowledge, this new set is significantly larger than any previous effort. We describe these changes in the following text:

Prior work exploring large-scale predictions of antibody structures have provided insight into the structural commonalities across individuals, and provide evidence of a public structural repertoire (50). Further, comparison on the basis of structure (rather than sequence alone) has enabled discovery of convergent binders that diverge significantly in sequence (51).

The first set consists of 104,994 paired antibody sequences (clustered at 95% sequence identity) from the OAS database (34). These sequences are made up

of 35,731 human, 16,356 mouse, and 52,907 rat antibodies. The second set contains 1,340,180 unique paired human antibody sequences from the immune repertoires of four unrelated individuals (52). These sequences span the affinity maturation spectrum, consisting of both naive and memory B-cell sequences. The structures are predicted with low estimated RMSD by IgFold, indicating that they are accurate (Figure S14 and S15). We highlight the predicted accuracy of the CDR H3 loops for the 1.3M human antibody sequences in Figure 4G. The median length and predicted RMSD for this set are 13 residues and 1.95 Å, respectively. We note that the predicted RMSD values tend to be underestimations, and in practice the actual H3 loop RMSDs, were structures to be experimentally determined, would likely be higher. As of October 2022, only 2,448 unique paired antibody structures have been determined experimentally (33), and thus our predicted dataset represents an over 500-fold expansion of antibody structural space.

In the abstract, the authors make the following statement: “Accurate structure prediction on this timescale makes possible avenues of investigation that were previously infeasible”. However, in the paper they show that IgFold has about equal speed and around a 10% increase in accuracy with respect to an already published method ABlooper. This statement needs to be removed or rephrased to say equally fast but improved performance.

We have now expanded our analysis to measure the runtimes of all methods with publicly available code on identical hardware. This analysis shows that IgFold provides significant speed improvements vs alternative methods in the full-atom prediction setting (including ABlooper), which we believe is the most relevant application of such models. We have added the following text to discuss these findings:

Although the performance of the deep learning methods for antibody structure prediction is largely comparable, the speed of prediction is not. Grafting-based methods, such as RepertoireBuilder, tend to be much faster than deep learning methods (if a suitable template can be found). However, as reported above, this speed is obtained at the expense of accuracy. Recent deep learning methods for antibody structure prediction, including DeepAb, ABlooper, and NanoNet, have claimed faster prediction of antibody structures as compared to general methods like AlphaFold. For our benchmark, all deep learning methods were run on identical hardware (12-core CPU with one A100 GPU), allowing us to directly compare their runtimes. All computed runtimes are measured from sequence to full-atom structure, using the recommended full-atom refinement protocols for each method. We could not evaluate the runtimes of RepertoireBuilder as no code has been published. The results of this comparison are summarized in Figure 4E-F.

For paired antibodies, we find that IgFold is significantly faster any other method tested. On average, IgFold takes 22 seconds to predict a full-atom structure from sequence. The next fastest method, ABlooper, averages nearly three minutes (174 seconds) for full-atom structure prediction. Although ABlooper rapidly predicts coordinates in an end-to-end fashion, the outputs require expensive refinement in OpenMM to correct for geometric abnormalities and add side chains. The ColabFold (12) implementation of AlphaFold-Multimer evaluated here averages just over seven minutes (435 seconds) on average for full-atom structure prediction. This is considerably faster than the original implementation of AlphaFold-Multimer, which required an expensive MSA search and repeated model compilation for every prediction. Finally, the slowest method for paired antibody structure prediction was DeepAb, which averaged over twelve minutes (750 seconds). DeepAb is considerably slower by design, as it requires minimization of predicted inter-residue potentials in Rosetta. We also investigated the impact of sequence length on prediction times. In general, the runtimes of all methods increased with sequence length (Figure S13A). DeepAb and ABlooper were the most sensitive to sequence length, with AlphaFold-Multimer and IgFold scaling more favorably.

For nanobodies, we again find that IgFold outpaces alternative methods for full-atom structure prediction, requiring an average of 11 seconds. NanoNet was the next fastest method, averaging 15 seconds for full-atom structure prediction. Similar to ABlooper for paired antibodies, NanoNet outputs require expensive refinement to correct for unrealistic backbone geometries and add side chains. DeepAb was able to predict nanobody structures in just under four minutes (224 seconds) on average. Finally, the slowest method for nanobody structure prediction was AlphaFold, which averaged nearly six minutes (345 seconds). As with paired antibodies, we also investigated the impact of sequence length on prediction times. In general, the runtimes of all methods increased with sequence length (Figure S13B). Although NanoNet had several outlier cases that required significant refinement, the prediction times for a majority of targets increased with sequence length. We also note that for methods capable of predicting both nanobody and paired antibody structures, runtimes tend to roughly double in the paired setting (scaling linearly with total length), as expected.

Why is AlphaFold2 used to generate the augmented structural dataset and AlphaFold Multimer for the benchmark?

We have added the following text to the manuscript to clarify that we used a modified AlphaFold to predict paired antibodies because AlphaFold-Multimer was not yet released:

Because AlphaFold-Multimer was not yet released, all predictions were performed with the original AlphaFold model.

The template data is successfully incorporated into predictions section is confusing. Is it surprising that given the true values for certain parts of the structure the model will be capable of predicting these accurately?

We agree with the authors that it is not surprising that the model can successfully incorporate the true structure (in the form of templates) into its predictions when provided. We have added the following text to provide context for highlighting this result:

Although these results are not surprising, they showcase a key functionality lacking in prior antibody-specific methods.

For nanobody structure prediction the authors should compare to Nanonet (T. Cohen, 2021) which claims higher accuracy than AlphaFold2 and greater speed than that claimed in this paper.

We have now added a comparison to NanoNet on our expanded nanobody structure benchmark. Notably, we do not find improvements over AlphaFold2 by NanoNet, and find that the prediction speed of NanoNet (for a full-atom structure, rather than unrefined backbone coordinates) is comparable to or slower than IgFold on average (see response to earlier comment). The following text has been added to introduce NanoNet and describe its performance on the benchmark:

Another tool, NanoNet, has been trained specifically for prediction of single-chain antibodies (nanobodies) and provides fast predictions.

NanoNet, trained specifically for nanobody structure prediction, outperforms DeepAb (average RMSD of 5.43 Å).

The authors claim that homology modelling is less accurate for the framework (line 25). However their results show otherwise (AbodyBuilder the homology modelling methods that is compared to is more accurate at predicting the light chain framework than IgFold in the used benchmark).

Indeed, as the reviewer notes, the benchmark data presented do not support this claim. We have adjusted the text to remove this claim, and instead note that ABlooper relies on external tools for framework modeling.

ABlooper, while faster and more informative, relies on external tools for framework modeling, cannot incorporate CDR loop templates, and does not support nanobody modeling.

In line 312 the authors claim that IgFold approaches the accuracy of AlphaFold2 on nanobodies. From the results it appears that AlphaFold is still a far more accurate on these proteins.

We have now expanded our nanobody benchmark using the same methodology described in our submission to 76 structures (from 23) using new structures deposited in the PDB. On the expanded benchmark, we find that the difference in performance between AlphaFold2 and IgFold is significantly reduced compared to our initial, smaller benchmark. These results are described in the following additions:

As with paired antibodies, all methods evaluated produced highly accurate predictions for the framework residues, with the average RMSD ranging from 0.57 Å to 0.80 Å. No method achieves sub-angstrom accuracy on average for CDR1 loops, though AlphaFold and IgFold achieve the best performance. For CDR2 loops, we observe a substantial improvement by IgFold and the other deep learning methods over RepertoireBuilder, with AlphaFold achieving the highest accuracy on average. For the CDR3 loop, RepertoireBuilder prediction quality is highly variable (average $\text{RMSD}_{\text{CDR3}}$ of 7.54 Å), reflective of the increased difficulty of identifying suitable template structures for the long, conformationally diverse loops. DeepAb achieves the worst performance for CDR3 loops, with an average $\text{RMSD}_{\text{CDR3}}$ of 8.52 Å, probably because its training dataset was limited to paired antibodies (14), and thus the model has never observed the full range of conformations accessible to nanobody CDR3 loops. NanoNet, trained specifically for nanobody structure prediction, outperforms DeepAb (average $\text{RMSD}_{\text{CDR3}}$ of 5.43 Å). AlphaFold displays the best performance for CDR3 loops, with an average $\text{RMSD}_{\text{CDR3}}$ of 4.00 Å, consistent with its high accuracy on general protein sequences. IgFold CDR3 predictions tend to be slightly less accurate than those of AlphaFold (average $\text{RMSD}_{\text{CDR3}}$ of 4.25 Å), but are significantly faster to produce (eleven seconds for IgFold, versus six minutes for the ColabFold implementation of AlphaFold).

To better understand the distinctions between IgFold- and AlphaFold-predicted nanobody structures, we highlight two examples from the benchmark. First, we compared the structures predicted by both methods for the benchmark target 7AQZ (unpublished, Figure 2F). This nanobody features a 15-residue CDR3 loop that adopts the "stretched-twist" conformation (44), in which the CDR3 loop bends to contact the framework residues that would otherwise be obstructed by a light chain in a paired antibody. IgFold correctly predicts this nanobody-specific loop conformation ($\text{RMSD}_{\text{CDR3}} = 2.81 \text{ \AA}$), while AlphaFold predicts an extended CDR3 conformation ($\text{RMSD}_{\text{CDR3}} = 7.08 \text{ \AA}$). Indeed, there are other cases where either IgFold or AlphaFold correctly predicts the CDR3

loop conformation while the other fails (see off-diagonal points in Figure S8G). In the majority of such cases, AlphaFold predicts the correct conformation, yielding the lower average CDR3 RMSD. In a second example, we compared the structures predicted by both methods for the benchmark target 7AR0 (unpublished, Figure 2G). This nanobody has a long 17-residue CDR3 loop with a short helical region. Although both methods correctly predict the loop conformation, IgFold fails to predict the helical secondary structure, resulting in a less accurate prediction (RMSDCDR3 = 2.27 Å) than that of AlphaFold (RMSDCDR3 = 0.84 Å). Such structured loops highlight a key strength of AlphaFold, which was trained on a large dataset of general proteins and has thus encountered a broad variety of structural arrangements, over IgFold, which has observed relatively few such structures within its training dataset.

The code is provided under a non-commercial licence.

Minor comments:

When showing clusters generated from AntiBERTy embeddings for each CDR, the authors argue that the clustering plots show that the encodings include structural features. However, the claim that the organization observed within the embedding space corresponds to different canonical folds is debatable. Furthermore, for CDR-H3 sequences are being labelled based on CDR length.

In response to the reviewer's comment, we have softened the claim that AntiBERTy has learned structural features of antibodies from sequence pre-training. We now note that there is some organization in the dimensionality reduction analysis, but we only claim that the model has picked up on some distinguishing features. We also note that in the absence of defined clusters, CDR H3 loops are labeled according to length. We note this in the text and have now added this to the supplemental figure legend as well. These changes are described below:

To determine whether the CDR loop representations encoded structural features, we labeled each point according to its canonical structural cluster. For CDR H3, which lacks canonical clusters, we instead labeled by loop length. **For the five CDR loops that adopt canonical folds we observed some organization within the embedded space, particularly for CDR1 loops.** For the CDR H3 loop, we found that the embedding space did not separate into natural clusters, but was rather organized roughly in accordance with loop length. **These results suggest that AntiBERTy has learned some distinguishing structural features of CDR loops through sequence pre-training alone.**

Updated supplemental figure legend:

For CDR H3, points are labeled according to loop length, **as canonical structures are not defined.**

In line 269, the authors claim that their predicted RMSD is well-calibrated with respect to the real RMSD. Although they show that both these metrics do correlate, in every plot where they compare these values the predicted RMSD consistently predicts lower values than the true RMSD. The authors should at least comment on this.

We have added some discussion of the underestimation of CDR RMSD and suggested potential sources of this behavior. We have also reworded our statement claiming calibration on unnatural sequences to instead note that the accuracy estimation is sensitive to such out-of-distribution sequences. These changes, along with updates reflecting new results given our larger benchmark, are provided below:

We observed significant correlations between the predicted error and the loop RMSDs from native for all the paired Fv CDR loops (Figure S10). For CDR H2 and CDR L2 loops, the correlations between predicted and measured RMSD were notably weaker. However, given the relatively high accuracy of predictions for these loops, there was little error to detect. For nanobodies, we observed significant correlations between the predicted error and RMSD for all the CDR loops (Figure S11). Interestingly, for all loops the model tended to predict lower RMSD than was measured. This may be a result of the imbalance between the smaller number of residues with higher RMSD (CDR loops) and the greater number with lower RMSD (framework residues). In the future, this miscalibration may be solved by using a weighted loss function that penalizes larger errors more heavily. However, the model's ability to effectively rank the accuracy of different CDR loops is still useful for identifying potentially inaccurate predictions.

This suggests that the RMSD predictions from IgFold are **sensitive** to unnatural antibody sequences and should be informative for a broad range of antibody structure predictions.

When showing the predicted RMSD for the generated structural OAS dataset, it would be beneficial to split this by CDR, as the whole Fv RMSD is often uninformative.

We have now performed a more substantive investigation of the properties of our predicted antibody datasets. We show the relationship between CDR loop lengths and predicted RMSD, as well as plot univariate marginals to show the distributions of these parameters individually. We provide these results for both predicted datasets in new supplemental figures S14 and S15.

The primary advantage of IgFold over other highly accurate methods like AlphaFold is its speed at predicting antibody structures. This speed enables large-scale prediction of antibody structures on modest compute resources. Prior work exploring large-scale predictions of antibody structures have provided insight into the structural commonalities across individuals, and provide evidence of a public structural repertoire (50). Further, comparison on the basis of structure (rather than sequence alone) has enabled discovery of convergent binders that diverge significantly in sequence (51). To demonstrate the utility of IgFold's speed for such analyses, we predicted structures for two non-redundant sets of paired antibodies. The first set consists of 104,994 paired antibody sequences (clustered at 95% sequence identity) from the OAS database (34). These sequences are made up of 35,731 human, 16,356 mouse, and 52,907 rat antibodies. The second set contains 1,340,180 unique paired human antibody sequences from the immune repertoires of four unrelated individuals (52). These sequences span the affinity maturation spectrum, consisting of both naive and memory B-cell sequences. The structures are predicted with low estimated RMSD by IgFold, indicating that they are accurate (Figure S14 and S15). We highlight the predicted accuracy of the CDR H3 loops for the 1.3M human antibody sequences in Figure 4G. The median length and predicted RMSD for this set are 13 residues and 1.95 Å, respectively. We note that the predicted RMSD values tend to be underestimations, and in practice the actual H3 loop RMSDs, were structures to be experimentally determined, would likely be higher. As of October 2022, only 2,448 unique paired antibody structures have been determined experimentally (33), and thus our predicted dataset represents an over 500-fold expansion of antibody structural space. These structures are made available for use in future studies.

There is a typo in the caption of Figure 4. It should be structure instead of structre.

Thank you. This and similar typos have been corrected.

Reviewer 2

Summary:

The paper presents IgFold, a deep learning method for antibody structure prediction. IgFold is an end-to-end neural network that consists of a pre-trained language model trained on 558M natural antibody sequences, followed by graph networks that directly predict backbone atom coordinates. The network allows for ingestion of template information and provides uncertainty estimates in addition to the residue locations. IgFold performance was compared with four baseline models. The results show that the accuracy is lower than AlphaFold-Multimer, similar to DeepAb, and better than two non-

deep learning baselines. The authors mention that IgFold provides faster predictions than AlphaFold (about 50 times faster) and DeepAb (about 10 times faster) and is comparable in time to the other two baselines.

Technical contribution:

The proposed neural network combines good elements of previously proposed networks such as AlphaFold. Thus, the paper does not have novelty with respect to advancing state of the art in deep learning. However, it is a reasonable application of deep learning to this particular bioinformatics problem. The value of this paper should be found in its potential to advance science.

Since submitting our manuscript for publication, two new methods (ESMFold, OmegaFold) employing approaches similar to IgFold have been described in preprints. We have added the following text to emphasize our contribution in the introduction:

Our model was the first to combine a single-sequence pretrained language model with an equivariant structure module for protein structure prediction, an approach which has since seen success for general protein structure prediction.

Strengths:

The main strength of IgFold is that it provides good accuracy and fast predictions. So, it fills a gap between very accurate but slower AlphaFold and less accurate but faster ABodyBuilder. Thus, IgFold can be helpful in specific applications where gains in computation efficiency justify losses in accuracy.

In response to other reviewer comments, we have significantly expanded our benchmark with structures released since our submission. On the new benchmark, we find that IgFold performs equally well as AlphaFold on paired antibodies (a major target for drug development) and only slightly worse on nanobodies. We believe the speed difference will make IgFold a preferred tool for high-throughput antibody structure screens and analyses.

Results on the updated paired antibody benchmark are summarized in an updated **Table 1** and **Figure 2A**.

Results on the updated nanobody benchmark are summarized in an updated **Table 2** and **Figure 2E**, and described below:

AlphaFold displays the best performance for CDR3 loops, with an average $\text{RMSD}_{\text{CDR3}}$ of 4.00 Å, consistent with its high accuracy on general protein sequences. IgFold CDR3 predictions tend to be slightly less accurate than those

of AlphaFold (average $\text{RMSD}_{\text{CDR3}}$ of 4.25 Å), but are significantly faster to produce (eleven seconds for IgFold, versus six minutes for the ColabFold implementation of AlphaFold).

Presentation is very good. The paper is well organized, and everything is explained sufficiently well. Analysis of predictions on specific antibodies provides an insight into the performance of IgFold and baselines.

We thank the reviewer.

Sharing over 100k predictions with the research community is welcome

Thank you. We hope the expanded set of structures described above will be of use to researchers.

Weaknesses and Suggestions:

Main weakness of the work is the lower accuracy of IgFold compared to AlphaFold. To me, this is the paper's most interesting result: AlphaFold, trained on a large set of structures without an attempt to specialize in antibody structures, is more accurate than IgFold, explicitly trained to do well on antibodies. Such a result is not unheard of in deep learning because foundational models in other domains (e.g., in NLP and computer vision) also often do better in specialized applications than the smaller models made specifically for those specialized applications. [Suggestion] It will be helpful for the authors to point to that similarity with other domains.

As noted above, we have significantly expanded our benchmark with structures released since our submission. On the new benchmark, we find that IgFold performs equally well as AlphaFold on paired antibodies and only slightly worse on nanobodies.

Results on the updated paired antibody benchmark are summarized in an updated Table 1 and Figure 2A.

Results on the updated nanobody benchmark are summarized in an updated Table 2 and Figure 2E, and described below:

AlphaFold displays the best performance for CDR3 loops, with an average $\text{RMSD}_{\text{CDR3}}$ of 4.00 Å, consistent with its high accuracy on general protein sequences. IgFold CDR3 predictions tend to be slightly less accurate than those of AlphaFold (average $\text{RMSD}_{\text{CDR3}}$ of 4.25 Å), but are significantly faster to produce (eleven seconds for IgFold, versus six minutes for the ColabFold implementation of AlphaFold).

The authors attempt to justify IgFold by comparing prediction speed. Unfortunately, the computational time is mentioned only briefly in lines 172-180 and 217, which does not amount to a thorough analysis. [Suggestion 1] To make this point stronger, the authors should devote more space to this aspect. It would be good to see the prediction times of each model on each of the test structures and see how the sequence length impact the prediction times. [Suggestion 2] Because low prediction cost is stated as the objective of this paper, the authors should change how they present their work. It would be important to discuss how the authors designed IgFold with that objective in mind. It would be important to provide an ablation study that demonstrates (1) that reducing IgFold architecture comes at the cost of unacceptable deterioration in accuracy and (2) that additional layers do not lead to further improvements in accuracy. [Suggestion 3] It would be important for the authors to provide more detail about why AlphaFold and DeepAb are slower than IgFold. In particular, what are the main reasons the baselines are slower, and would removing or compressing some of their components cause significant loss of accuracy?

We have now expanded our analysis to measure the runtimes of all methods with publicly available code on identical hardware. This analysis shows that IgFold provides significant speed improvements vs alternative methods in the full-atom prediction setting, which we believe is the most relevant application of such models. We also show the impact of sequence length on prediction times in a new supplemental figure. We have added the following text to discuss these findings:

Although the performance of the deep learning methods for antibody structure prediction is largely comparable, the speed of prediction is not. Grafting-based methods, such as RepertoireBuilder, tend to be much faster than deep learning methods (if a suitable template can be found). However, as reported above, this speed is obtained at the expense of accuracy. Recent deep learning methods for antibody structure prediction, including DeepAb, ABlooper, and NanoNet, have claimed faster prediction of antibody structures as compared to general methods like AlphaFold. For our benchmark, all deep learning methods were run on identical hardware (12-core CPU with one A100 GPU), allowing us to directly compare their runtimes. All computed runtimes are measured from sequence to full-atom structure, using the recommended full-atom refinement protocols for each method. We could not evaluate the runtimes of RepertoireBuilder as no code has been published. The results of this comparison are summarized in Figure 4E-F.

For paired antibodies, we find that IgFold is significantly faster any other method tested. On average, IgFold takes 22 seconds to predict a full-atom structure from sequence. The next fastest method, ABlooper, averages nearly three minutes (174 seconds) for full-atom structure prediction. Although ABlooper rapidly predicts coordinates in an end-to-end fashion, the outputs require

expensive refinement in OpenMM to correct for geometric abnormalities and add side chains. The ColabFold (12) implementation of AlphaFold-Multimer evaluated here averages just over seven minutes (435 seconds) on average for full-atom structure prediction. This is considerably faster than the original implementation of AlphaFold-Multimer, which required an expensive MSA search and repeated model compilation for every prediction. Finally, the slowest method for paired antibody structure prediction was DeepAb, which averaged over twelve minutes (750 seconds). DeepAb is considerably slower by design, as it requires minimization of predicted inter-residue potentials in Rosetta. We also investigated the impact of sequence length on prediction times. In general, the runtimes of all methods increased with sequence length (Figure S13A). DeepAb and ABlooper were the most sensitive to sequence length, with AlphaFold-Multimer and IgFold scaling more favorably.

For nanobodies, we again find that IgFold outpaces alternative methods for full-atom structure prediction, requiring an average of 11 seconds. NanoNet was the next fastest method, averaging 15 seconds for full-atom structure prediction. Similar to ABlooper for paired antibodies, NanoNet outputs require expensive refinement to correct for unrealistic backbone geometries and add side chains. DeepAb was able to predict nanobody structures in just under four minutes (224 seconds) on average. Finally, the slowest method for nanobody structure prediction was AlphaFold, which averaged nearly six minutes (345 seconds). As with paired antibodies, we also investigated the impact of sequence length on prediction times. In general, the runtimes of all methods increased with sequence length (Figure S13B). Although NanoNet had several outlier cases that required significant refinement, the prediction times for a majority of targets increased with sequence length. We also note that for methods capable of predicting both nanobody and paired antibody structures, runtimes tend to roughly double in the paired setting (scaling linearly with total length), as expected.

The authors repeatedly imply (lines 199, 227, 236) that IgFold predictions could be helpful despite being less accurate than AlphaFold. The argument seems to be that IgFold predictions improve the diversity of predicted structures, which could be helpful in some downstream applications. This claim is suspect because it is well known from the mixture of experts research that adding inferior predictors to an ensemble leads to lower accuracy. [Suggestion] For a claim of that sort to hold, the authors should show at least one example (hopefully, not cherry-picked) where adding IgFold to AlphaFold predictions is helpful.

We have removed prior statements throughout the results text where we claim that the diversity of predictions could be useful for downstream applications, and instead simply report on the finding.

The authors imply that IgFold finds a sweet spot between accuracy and prediction time.

However, they do not provide any specific practical example where IgFold would be desirable over more accurate or faster models. [Suggestion] The authors should introduce a cost function that combines accuracy and time to demonstrate it. For example, in what application 10% accuracy loss can be justified by 10 or 100 times lower computational cost?

On the updated benchmark (see response to earlier reviewer comment) we have found that there is little difference in average accuracy between IgFold and the AlphaFold models. We are hesitant to introduce such a cost function, as the accuracy demands placed on these models may vary widely based on application. For example, to evaluate hydrophobicity in framework structures, IgFold's speed should be incredibly valuable. Indeed, for paired antibodies, AlphaFold appears to offer no advantage over IgFold. Given the slight degradation of performance on nanobodies, we have removed our claims in the results text that the diversity of predictions may be useful in practice, as we believe the 30-fold speedup over AlphaFold is itself compelling.

On one hand, it is positive that the authors used the structures discovered after July 1, 2021, as test data in the evaluation. On another, the set of 67 paired antibodies and 21 nanobodies is too small to evaluate IgFold properly. [Suggestion] It would be important to create the second, larger test data set using structures prior to July 1, 2021. One suggestion is to start selecting test structures from the training data one by one at random and remove from the training data set all similar (e.g., 70% and higher identity in CDRs) sequences. In this way, the test sequences will be sufficiently different from the remaining training data and would allow proper testing without the fear of biased results.

We have now expanded our benchmark sets using the same methodology described in our original submission. Additions to the PDB have now allowed us to compare performance on 197 paired antibody structures and 71 nanobody structures. The overarching conclusions remain the same, though we do find that performance of IgFold on nanobodies is more comparable to AlphaFold2 than we initially presented. The updated text describing these results is provided in an earlier comment to the reviewer.

The results shown in Figures 3 and 4 and the accompanying discussion only refer to 4 specific structures. It is not clear why those structures are selected over other test structures. Without justification, a reader can start wondering if those structures are cherry-picked.

We have added additional text to the benchmarking results section to motivate the selection of the specific examples highlighted in Figure 3. Although there is nothing special about these particular examples, they illustrate the structural implications of a trend observed across the broader set of benchmark targets. In

the interest of space, and to focus more attention on other reviewer comments, we have removed the specific structural examples from Figure 4.

To illustrate the structural implications of these differences in predictions, we highlight two targets from the benchmark where IgFold and AlphaFold-Multimer diverge.

REVIEWER COMMENTS

Reviewer #1 (Remarks to the Author):

Thank the authors for addressing our comments but still have a few concerns.

1. Although it makes sense to compare with the ColabFold version of AlphaFold-Multimer and AlphaFold2 for high-throughput modelling, one would expect it to generate models of inferior quality to those generated using the original implementation as described in their papers. The authors should either compare against the implementations of these methods as described in the literature or refer to them as ColabFold throughout the paper and acknowledge the potential differences.
2. In the abstract, the authors make the following statement: "(...) we predicted structures for 1.4 million paired antibody sequences, expanding the observed antibody structural space by over 500 fold". Although this is undoubtedly a great contribution to the field, the way it is phrased makes it seem like IgFold predicted models are of equal value to experimentally resolved structures. I think this needs to be rephrased slightly.
3. 1. Certain parts of the text have been edited but not highlighted or mentioned in the response. For example, in the results section the authors use to compare against ABodyBuilder and now compare against RepertorieBuilder

Reviewer #2 (Remarks to the Author):

The revised version of the paper and the responses to the reviewers successfully address all the issues raised in the original reviews. Evaluation on the increased test set shows that the gap in accuracy between IgFold and AlphaFold is much smaller than in the original submission. New runtime experiments clarify the differences in predictions costs amonts different approaches. It is positive that the authors now provide a significantly expanded set of predicted structures. It is important to emphasize that the github repository with the code, trained predictor, and predicted structures is a key contribution of this work. It will be very important for the authors to maintain this repository after the paper is published.

Response to reviewers

We thank the reviewers for their comments and critiques, which have undoubtedly resulted in a stronger manuscript. Below we detail the changes made in response to the reviewer's comments. For convenience, the original comments are included, while our responses (indented) are below. Changes to the manuscript are indicated by **red text**.

Reviewer 1

Comments:

Although it makes sense to compare with the ColabFold version of AlphaFold-Multimer and AlphaFold2 for high-throughput modelling, one would expect it to generate models of inferior quality to those generated using the original implementation as described in their papers. The authors should either compare against the implementations of these methods as described in the literature or refer to them as ColabFold throughout the paper and acknowledge the potential differences.

Although speed and accuracy are commonly at odds, benchmarks and blind evaluations have demonstrated that ColabFold provides improvements to both over the original DeepMind AlphaFold pipeline. In the ColabFold publication (<https://www.nature.com/articles/s41592-022-01488-1>), the authors find considerable speed-ups with optimizations to the MSA generation and model inference steps, without degradations to performance. Further, at the recent CASP15 blind structure prediction assessment (https://predictioncenter.org/casp15/zscores_final.cgi), ColabFold outperformed the original AlphaFold pipeline from DeepMind, despite using the same model weights. Given these results, as well as the ubiquity of ColabFold amongst practitioners, we believe it is logical to compare to ColabFold instead of the DeepMind reference implementation.

To the reviewer's recommendation regarding attribution of results in our publication to AlphaFold vs ColabFold, we opted for the former because:

- (1) ColabFold is ultimately an alternative means of running AlphaFold and
- (2) After consulting the ColabFold corresponding author, Sergey Ovchinnikov, he agreed that the model performance should be attributed AlphaFold. Additionally, he suggested that recognition be given to the MMseqs2 method used to generate the input multiple-sequence alignments, as this is the primary distinction in the two pipelines.

To make this decision more explicit, and highlight the considerations above, we have added the following text at the beginning of the benchmarking results section:

We opted to benchmark the ColabFold (Mirdita et al., *Nature Methods* (2022)) implementation of AlphaFold, rather than the original pipeline from DeepMind, due to its significant runtime acceleration and similar accuracy.

And in the methods section:

The ColabFold pipeline utilizes the model weights trained by DeepMind, but replaces the time-consuming MSA generation step with a faster search via MMseqs2 (Steinegger et al., *Nature Biotechnology* (2017)).

In the abstract, the authors make the following statement: “(...) we predicted structures for 1.4 million paired antibody sequences, expanding the observed antibody structural space by over 500 fold”. Although this is undoubtedly a great contribution to the field, the way it is phrased makes it seem like IgFold predicted models are of equal value to experimentally resolved structures. I think this needs to be rephrased slightly.

We have adjusted our phrasing to emphasize that we believe our predicted structures will provide insights through future studies, but not suggest equivalency with experimentally determined structures:

As a demonstration of IgFold's capabilities, we predicted structures for 1.4 million paired antibody sequences, **providing structural insights to 500-fold more antibodies than have experimentally determined structures.**

Certain parts of the text have been edited but not highlighted or mentioned in the response. For example, in the results section the authors use to compare against ABodyBuilder and now compare against RepertoireBuilder.

We apologize for the oversight in failing to highlight these edits in the previous response.

To the reviewer's specific point about substituting ABodyBuilder with RepertoireBuilder in the latest version of the manuscript, we had to make this change to evaluate on the expanded benchmark requested by reviewers. In the period between revisions, the ABodyBuilder server was updated and no longer provided homology modeling as benchmarked here. Unfortunately, as the code for ABodyBuilder is not publicly available, we had to replace the method. RepertoireBuilder uses an alternative, though still grafting-based, approach to antibody modeling. In prior work, RepertoireBuilder has been shown to perform comparable to ABodyBuilder, making it a suitable substitution to continue benchmarking these types of methods (Ruffolo et al., *Patterns* (2022)).

Reviewer 2

The revised version of the paper and the responses to the reviewers successfully address all the issues raised in the original reviews. Evaluation on the increased test set shows that the gap in accuracy between IgFold and AlphaFold is much smaller than in the original submission. New runtime experiments clarify the differences in predictions costs amongs different approaches. It is positive that the authors now provide a significantly expanded set of predicted structures. It is important to emphasize that the github repository with the code, trained predictor, and predicted structures is a key contribution of this work. It will be very important for the authors to maintain this repository after the paper is published.

We thank the reviewer for their helpful feedback on the manuscript.

REVIEWERS' COMMENTS

Reviewer #1 (Remarks to the Author):

The authors have done an excellent job and the overall paper is well presented and explained.